# Increasingly negative tropical water–interannual CO$_2$ growth rate coupling

Laibao Liu[1]✉, Philippe Ciais[2], Mengxi Wu[3], Ryan S. Padrón[1], Pierre Friedlingstein[4], Jonas Schwaab[1], Lukas Gudmundsson[1] & Sonia I. Seneviratne[1]

Terrestrial ecosystems have taken up about 32% of the total anthropogenic CO$_2$ emissions in the past six decades[1]. Large uncertainties in terrestrial carbon–climate feedbacks, however, make it difficult to predict how the land carbon sink will respond to future climate change[2]. Interannual variations in the atmospheric CO$_2$ growth rate (CGR) are dominated by land–atmosphere carbon fluxes in the tropics, providing an opportunity to explore land carbon–climate interactions[3–6]. It is thought that variations in CGR are largely controlled by temperature[7–10] but there is also evidence for a tight coupling between water availability and CGR[11]. Here, we use a record of global atmospheric CO$_2$, terrestrial water storage and precipitation data to investigate changes in the interannual relationship between tropical land climate conditions and CGR under a changing climate. We find that the interannual relationship between tropical water availability and CGR became increasingly negative during 1989–2018 compared to 1960–1989. This could be related to spatiotemporal changes in tropical water availability anomalies driven by shifts in El Niño/Southern Oscillation teleconnections, including declining spatial compensatory water effects[9]. We also demonstrate that most state-of-the-art coupled Earth System and Land Surface models do not reproduce the intensifying water–carbon coupling. Our results indicate that tropical water availability is increasingly controlling the interannual variability of the terrestrial carbon cycle and modulating tropical terrestrial carbon–climate feedbacks.

The interannual variations (IAV) of the CO$_2$ growth rate (CGR) are found to be strongly correlated with El Niño/Southern Oscillation (ENSO)[12,13] (for example, $R = -0.55$, $P < 0.05$ in ref. 12, Pearson correlation coefficient), particularly with tropical temperature variations[7–9] (for example, $R = 0.7$, $P < 0.01$ in ref. 7), despite the lower IAV of tropical temperature than for other locations[14]. The historical IAV sensitivity of CGR to tropical temperature was further identified as an observational constraint that can significantly lower uncertainties in projected tropical carbon budgets[5]. Compared to tropical temperature, concurrent tropical precipitation is not well correlated with CGR[15,16] (for example, $R = -0.19$, $P > 0.1$ in ref. 16) but lagged tropical precipitation was shown to strongly explain the IAV of CGR or tropical net land carbon flux[7,17] (for example, $R = -0.5$, $P < 0.05$ in ref. 7), resulting in an ambiguous role of water availability in controlling CGR from a process perspective. Recently, the launch of twin satellites of the Gravity Recovery and Climate Experiment (GRACE) enabled the direct measurement of terrestrial water storage (WS) variability, and a subsequent analysis showed that it is tightly coupled to CGR[11] ($R = -0.85$, $P < 0.01$). However, in the context of climate change, it remains unclear whether the identified terrestrial climate–carbon coupling is constant over time or may vary subject to changes in climate forcers and mean climate.

Here, we investigate the changes in the interannual relationship between tropical land climate conditions and CGR over the past decades. To complement the shorter observational record of the GRACE satellites, we also use recently reconstructed long-term WS variability[18]. Furthermore, the 6-month lagged yearly precipitation (LagP) can approximate aggregated tropical WS IAV well and correlates with CGR IAV, emerging as another efficient proxy for tropical terrestrial water availability IAV (Methods). This also helps to explain why lagged precipitation correlated well with CGR in previous findings[7,17].

## Observed climate–carbon coupling

All variables are detrended at yearly time scale by removing the long-term linear trend, as we focus on the relationship in interannual variability. The years following the eruptions of Mount Agung (1963 and 1964), El Chichón (1982) and Mount Pinatubo (1991–1993) are also excluded from analyses to avoid perturbations of unusual carbon flux anomalies[19]. For the entire 1960–2018 period, CGR is significantly correlated with both tropical temperature ($R_{T,CGR} = 0.64$, $P < 0.01$, Pearson correlation coefficient) and tropical WS ($R_{WS,CGR} = -0.58$, $P < 0.01$) (Fig. 1a). The opposite sign in the two relationships suggests that hotter (positive temperature anomaly) and drier (negative WS anomaly)

[1]Institute for Atmospheric and Climate Science, ETH Zurich, Zurich, Switzerland. [2]Laboratoire des Sciences du Climat et de l'Environnement, CEA-CNRS-UVSQ, Université Paris Saclay, Gif-sur-Yvette, France. [3]Joint Institute for Regional Earth System Science and Engineering (JIFRESSE), University of California, Los Angeles, Los Angeles, CA, USA. [4]College of Engineering, Mathematics and Physical Sciences, University of Exeter, Exeter, UK. ✉e-mail: laibao.liu@env.ethz.ch

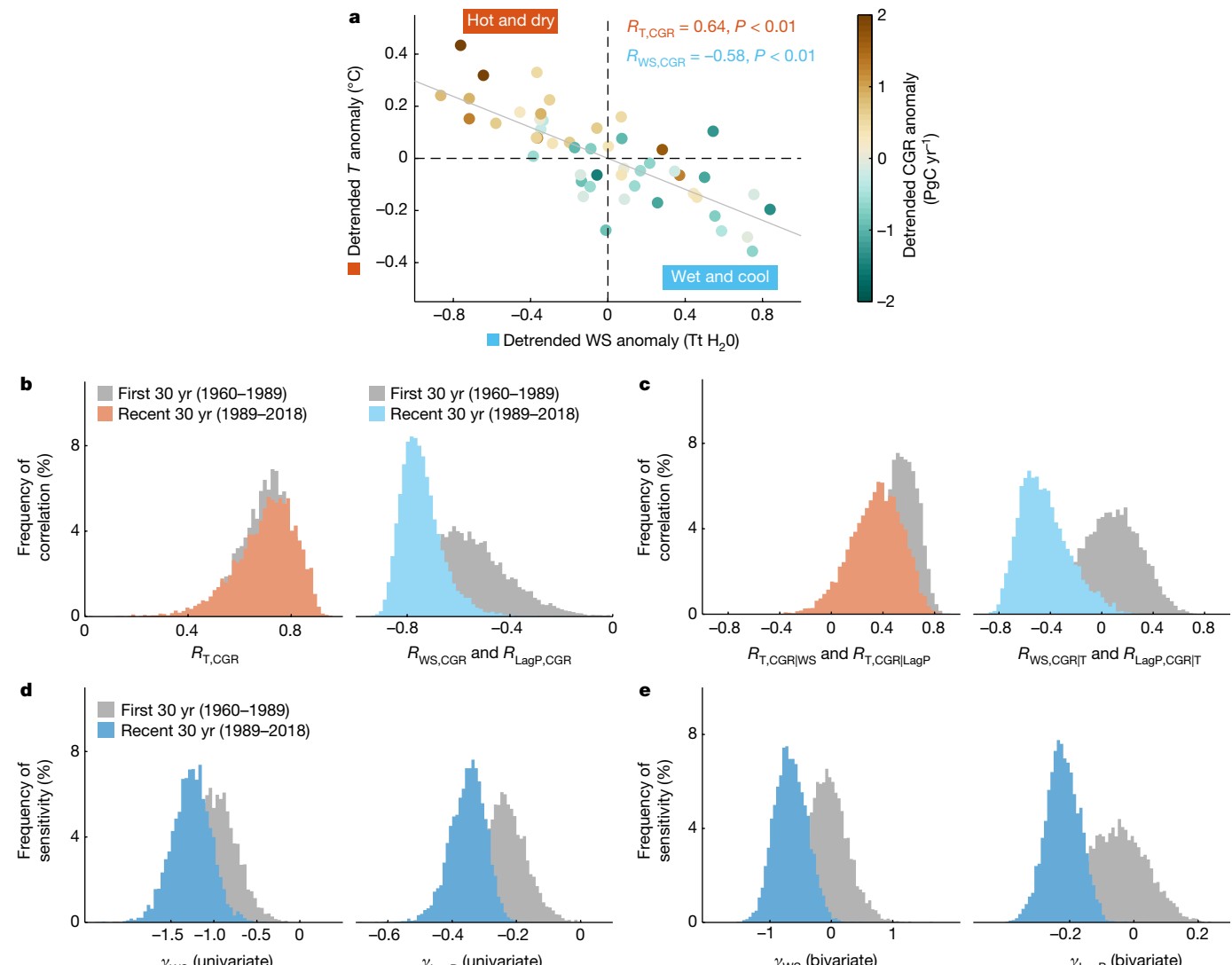

**Fig. 1 | Tropical land climate–carbon interannual relationships. a**, Yearly tropical temperature versus tropical WS versus CGR in detrended anomalies. The values of CGR are indicated by the colour bar. **b**, Histograms of climate–CGR interannual correlations in the first three decades (1960–1989) and in the more recent three decades (1989–2018), derived using 5,000 bootstrapping repeats. Both tropical WS and LagP are used to represent tropical water availability. **c**, As for **b** but showing histograms of the partial correlations of CGR to tropical temperature and tropical water after controlling tropical water and tropical temperature, respectively. **d**, Histograms of the interannual sensitivity of CGR to tropical WS ($\gamma_{WS}$) and LagP ($\gamma_{LagP}$) in the univariate regression for the same two periods, derived using 5,000 bootstrapping repeats. Unlike correlations, $\gamma_{WS}$ and $\gamma_{LagP}$ differ in magnitude because of the differences in WS and LagP IAV magnitude and are therefore shown separately. The unit of this sensitivity is PgC yr$^{-1}$ per Tt H$_2$O. **e**, As for **d** but showing $\gamma_{WS}$ and $\gamma_{LagP}$ estimated using the bivariate regression with both tropical water and tropical temperature as predictors. Ridge regression is used here to reduce biases from high collinearity between water and temperature (Methods).

climate conditions generally dampened the land carbon sink and thus enhanced atmospheric CO$_2$ growth in the past decades. There is also a small proportion of CGR that does not fit the general pattern, suggesting the role of other factors, such as exceptional (nonlinear) anthropogenic emissions or ocean carbon sink.

Next, we investigate how the correlation between climate and CGR changes from the first 30 years (1960–1989) to the most recent 30 years (1989–2018). The uncertainty of the correlation is quantified through bootstrapping (5,000 replications). The results show that the bootstrap distributions of temperature–CGR correlations are similar between the two periods but water–CGR correlations significantly become more negative over time ($P < 0.1$ averaged over correlation metrics, Fig. 1b and Extended Data Table 1). To check whether this increasingly negative water–carbon correlation is influenced by the possible confounding water–temperature coupling, we look at the temporal dynamic of water–temperature correlations and find that they are stable over time

(Supplementary Fig. 1). Partial correlations remove water–temperature correlations and their relative changes directly help to confirm that the increasingly negative tropical water–carbon correlation remains robust (Fig. 1c). We note that because terrestrial water variability can also indirectly influence the land carbon cycle by triggering atmospheric temperature extremes through the well-documented soil moisture–atmosphere feedbacks[20,21], it could be inappropriate to interpret partial correlation $R_{W,CGR|T}$ as total water impacts on CGR in the two periods, but the temporal changes of $R_{W,CGR|T}$ are useful here (Methods). In addition, we compute climate–CGR correlations for a 25-year moving window to provide insights into the gradual changes, which are quite smooth over time (Supplementary Fig. 2). To further test the robustness of changes in the interannual correlations between tropical water and CGR, we also consider alternative observation datasets of tropical yearly precipitation and tropical temperature (Extended Data Fig. 1a). To verify that the IAV in CGR does not primarily originate from fossil fuel emissions,

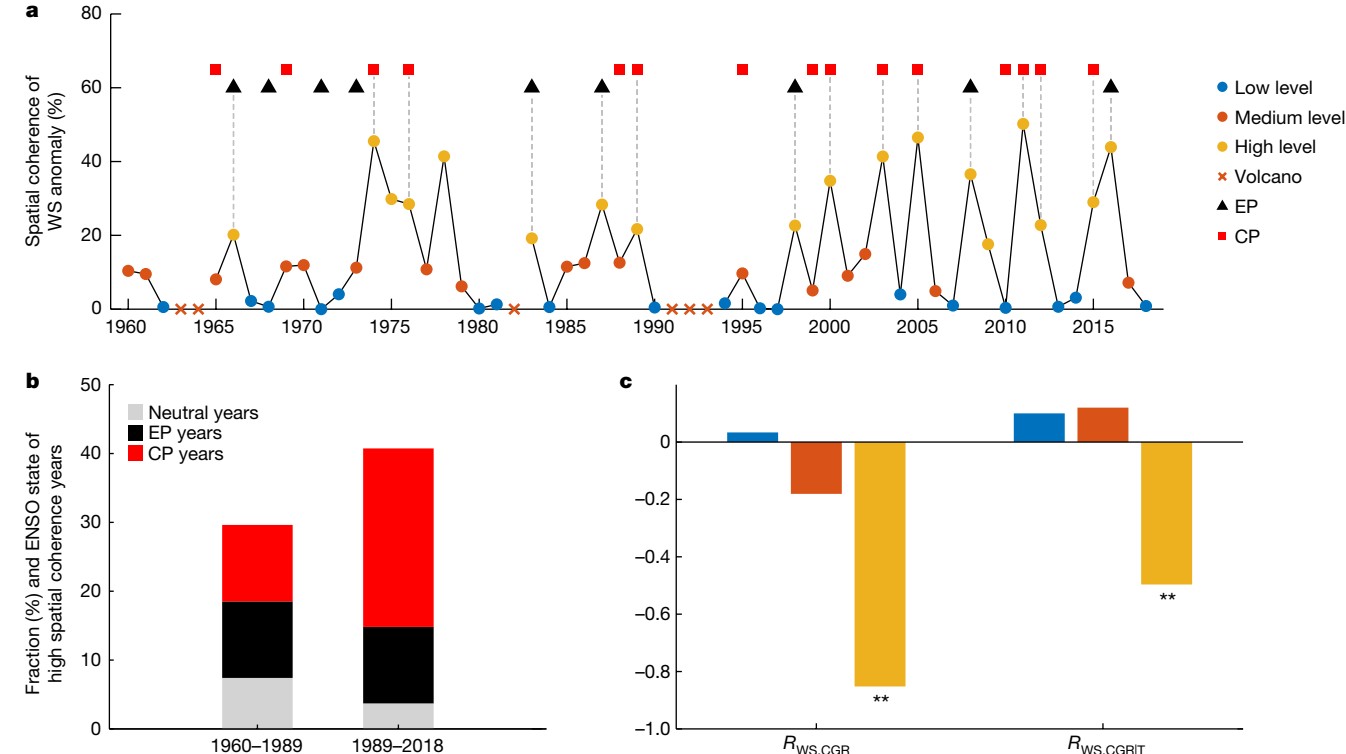

**Fig. 2 | Influence of ENSO on tropical land water–carbon coupling.**
**a**, Year-to-year variations of spatial coherence of tropical WS anomaly and ENSO. All years are classified into three subsets according to the level of spatial coherence: low level (0th to 33.3th percentile); medium level (33.3th to 66.6th percentile); and high level (66.6th to 100th percentile). Year is considered Eastern Pacific (EP) ENSO when the largest DJF SST anomaly over the region of 2° S–2° N, 110° E–90° W lies in the Eastern Pacific (east of 150° W) and Nino3 index exceeds 1 s.d. Year is considered Central Pacific (CP) ENSO when the corresponding largest DJF SST anomaly lies in the Central Pacific (west of 150° W) and Nino4 index exceeds 1 s.d. Volcano years are excluded from analyses. Grey vertical dashed lines connect the symbols of high spatial coherence and ENSO. **b**, Fraction of years with high spatial coherence within the first 30-yr period (1960–1989) and within the recent 30-yr period (1989–2018). Neutral years are identified as years not in the EP ENSO or CP ENSO state. **c**, Dependence of $R_{WS,CGR}$ and $R_{WS,CGR|T}$ on the spatial coherence of WS anomaly. **P < 0.05 (significant correlation).

land-use change and ocean uptake, we also use the residual land sink (RLS) instead of CGR (Extended Data Fig. 1b). These results all point to an increasingly negative correlation between tropical water and CGR on the interannual scale over the past decades.

The sensitivity of CGR IAV to tropical water is further estimated using linear regressions, defined as the slope of the regression between CGR and climate, with both variables detrended. In alignment with correlations, we first perform a univariate regression in which CGR is the function of tropical water alone to avoid possible underestimations of the sensitivity of CGR to tropical water variations. The univariate sensitivity of CGR to tropical WS and LagP increased (more negative) by about 35% on average from the previous 30 years (1960–1989) to the more recent 30 years (1989–2018) (Fig. 1d and Extended Data Table 1). For instance, the sensitivity to tropical WS increased from −0.95 ± 0.27 PgC yr⁻¹ per Tt H₂O (teratonnes of water) to −1.26 ± 0.23 PgC yr⁻¹ per Tt H₂O during the same two periods. We also perform bivariate linear regression with both tropical temperature and tropical water as predictors, and the sensitivity of CGR to tropical water variations is not significantly different from 0 in the first 30 years but becomes significantly negative in the recent 30 years ($P < 0.05$) (Fig. 1e). We use the ridge regression to reduce the effects of high collinearity between water and temperature on sensitivity estimates, but the ordinary least squares (OLS) regression also yields increasingly negative sensitivity over time (Extended Data Table 1). In addition, results based on a moving 25-yr window show that the sensitivity stalls within the recent 34-yr period, that is, after the time window centred on 1997 (1985–2009) (Supplementary Fig. 3). These results together suggest that the interannual relationship between tropical water and

CGR has become increasingly negative in the recent past (1989–2018) compared to previous climate conditions (1960–1989). Also, we find that the amplitude of temperature sensitivity enhancement from the bivariate regression is smaller than that previously reported[16,22], if WS or LagP, rather than concurrent precipitation, is used as the proxy for water availability (Extended Data Fig. 2). This is because the accurate proxy for water availability was not identified and available (that is, using concurrent precipitation (reported $R_{Pre,CGR} = −0.19$, $P > 0.1$ in ref. 16) rather than WS or LagP as the proxy starting from 1960 ($R_{LagP,CGR} = −0.68$, $P < 0.01$; Extended Data Fig. 3b)). Moreover, we extend the analysis time period by including 2011 to 2018 and observe a recent declining temperature sensitivity. All these results emphasize that it is crucial to integrate water availability into the carbon–climate feedback metric for better estimating climate-driven changes in tropical terrestrial carbon sink.

## ENSO teleconnections

Interannual changes in tropical climate are largely driven by ENSO[23]. Under anthropogenic warming, growing observational evidence shows robust changes in ENSO characteristics, such as increases in frequency and variability with shifts in types[24], especially the largest sea surface temperature (SST) anomaly shifts from the Eastern Pacific (EP) to the Central Pacific (CP) since the 1990s[25–28]. As a result, it is likely to alter the patterns of moisture and heat fluxes over tropical continents, for instance, triggering more extreme droughts and fires[29] and thus modulating terrestrial carbon–climate feedbacks. Indeed, we find that most years with high spatial coherence of tropical WS anomalies are ENSO

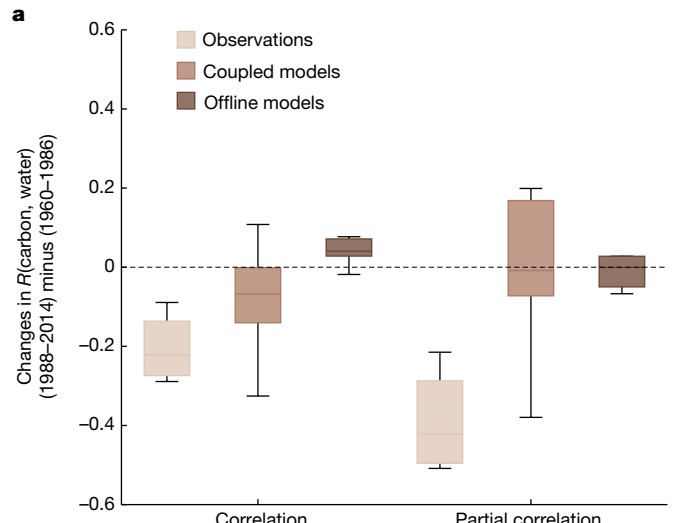

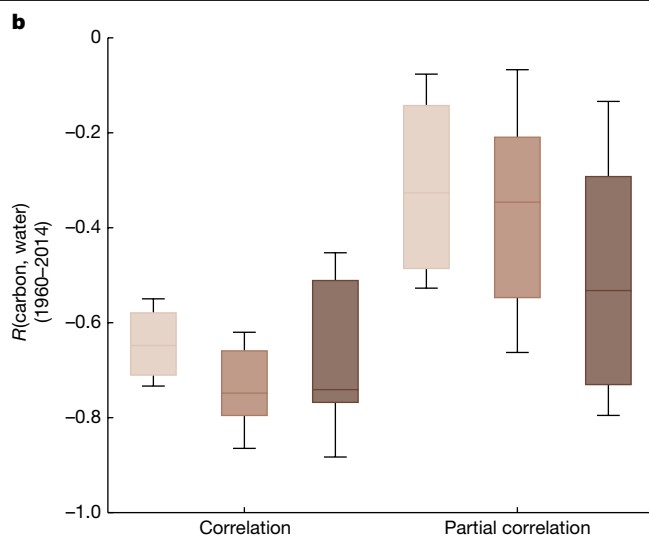

**Fig. 3 | Historical tropical land water–carbon interannual correlations in observations and models. a**, Changes in interannual correlations and partial correlations between tropical water availability and land carbon fluxes from the previous 27 years (1960–1986) to the recent 27 years (1988–2014). For partial correlations, tropical temperature is controlled. All variables are detrended at yearly scale in each corresponding window. For observations, CGR/RLS, reconstructed tropical WS/tropical LagP and tropical temperature are used for computations ($n = 4$). For models, global NEE, tropical total soil

moisture and tropical temperature from each model are used for computations (coupled models, $n = 9$; offline models, $n = 6$). Box plots show the distribution of estimates for all models, solid horizontal lines indicate the median values, boxes cover the interquartile range, and vertical lines reach the 5th and 95th percentiles. **b**, Same as **a** but showing the interannual correlations and partial correlations between tropical water availability and land carbon fluxes during 1960–2014.

years (Fig. 2a,b), for which spatial coherence is quantified by applying a metric adjusted from ref. 9 (Methods). There is still a small proportion of high spatial coherence that cannot be explained by ENSO, suggesting a role for other factors, such as the tropical Atlantic variability[30,31] and the Indian Ocean Dipole[32,33]. Compared to ENSO-neutral years, spatial patterns of WS anomaly during ENSO years are quite uniform (Extended Data Fig. 4). Then, from the first 30 years (1960–1989) to the most recent 30 years (1989–2018), the fraction of years at high spatial coherence levels increases from 30% to 41% because of increased contributions of CP ENSO and fewer neutral years (Fig. 2b). These results could be important because the dominant role of water availability in controlling carbon fluxes at a larger scale is affected by the spatial compensatory degree of water anomalies: water availability clearly dominates carbon flux IAV locally but this control could partly be spatially counterbalanced when aggregated[9]. To further confirm the effects of spatial coherence on tropical water–CGR coupling, we perform a new subset analysis. We first detrend all years of data by removing the long-term trend and then bin them into three subsets according to spatial coherence levels. The $R_{WS,CGR}$ is highly negative only when the spatial coherence is high (Fig. 2c). The $R_{WS,CGR|T}$ confirms that this dependence of water–carbon coupling on spatial coherence is not influenced by confounding temperature effects. Similar results are found when we replace WS with LagP (Supplementary Fig. 4). The sensitivity of CGR to WS is also expected to become apparently negative only when the spatial coherence is high (Supplementary Fig. 5). Another subset analysis using a 25-year moving window also supports the idea that the enhanced spatial coherence strengthens the increasingly negative tropical water–CGR coupling (Supplementary Fig. 6). These results indicate that identified ENSO-driven enhancement of spatial coherence over time is a likely explanation for the increasingly negative coupling between tropical water and CGR IAV. In the future, if CP ENSO events become more common with global warming, as predicted[34], tropical water might continue to increasingly control tropical terrestrial sink IAV. Further investigations on different impacts of EP ENSO and CP ENSO events on the terrestrial carbon cycle are useful to better understand future carbon–climate feedbacks[35,36].

## Diagnosis of CMIP6 models

The observational relationships between tropical climate and CGR are valuable metrics to diagnose the ability of models to simulate terrestrial climate–carbon interannual variability. We thus investigate whether state-of-the-art models participating in the Coupled Model Intercomparison Project Phase 6 (CMIP6) can capture this observed feature. We calculate the interannual correlation between tropical WS (using tropical total soil moisture as proxy) and simulated global net ecosystem exchange (NEE, ecosystem respiration minus photosynthesis) estimated by an ensemble of nine coupled Earth System Models (ESMs) and six offline Land Surface Models (LSMs) in the historical 1960–2014 period (Methods). We find that the (partial) correlations between simulated tropical soil moisture and global NEE are persistently high and thus remain almost unchanged over time in most ESMs and LSMs (Fig. 3a and Extended Data Fig. 5). Also, most models do not reproduce the increased sensitivity of NEE to tropical soil moisture, even though they differ largely in the absolute magnitude of the sensitivity (Extended Data Fig. 6). These results indicate that models do not capture the observed emerging enhancement of tropical water–carbon coupling over time, although models roughly capture the sign and strength of this interannual water–carbon relationship during 1960–2014 (Fig. 3b). The ability of models to reproduce observed tropical water–carbon coupling depends not only on the simulation of terrestrial water availability but also on process representations of the response of the carbon cycle to climate. We further find that models might not represent the latter part well because the modelled water–carbon coupling is stable over time, regardless of the large differences among simulated soil moisture. Specifically, the dominant spatial patterns of simulated soil moisture anomalies are largely divergent among models (Supplementary Figs. 7 and 8), although simulated soil moisture from all offline models can generally capture ENSO teleconnections (Supplementary Fig. 9). For coupled ESMs, the underpinning reason is more complex, for instance, they have known issues in simulating the probability of occurrence of historical EP ENSO and CP ENSO[37]. Compared to ecosystem respiration,

the simulated response of ecosystem gross primary production to soil moisture in models is more consistent (Supplementary Figs. 10 and 11). Further including possible modelled carbon fluxes from fire and other disturbances, that is, replacing NEE with net biome production, cannot help to explain the failure of models to reproduce the intensified water–carbon coupling (Supplementary Fig. 12). Models also possibly lack some critical process representations[11], such as the parameterization of deep water uptake[38], tree mortality[39] and plant root adaptation-related characteristics[40]. Therefore, the ability of models to project future terrestrial carbon–climate feedbacks are subject to uncertainties and these results call for improvements on water–carbon interactions to better constrain projections.

Direct observations of tropical net and constituent carbon fluxes covering such a long time period are lacking, limiting spatially explicit attributions of changes in carbon fluxes to specific regions and underpinning drivers. Recent evidence shows that aboveground carbon (AGC) fluxes in tropical semi-arid biomes are strongly associated with CGR IAV during 2011–2017[41]. Tropical AGC dynamics are retrieved from microwave satellite observations of vegetation optical depth (VOD). However, new evidence raises a caution about interpreting IAV of VOD-derived AGC ($AGC_{VOD}$) as biomass IAV alone because this might be more directly linked to soil moisture[42]. Therefore, although we find increased semi-arid $CGR$-$AGC_{VOD}$ coupling in the recent 15 years (2002–2016) relative to the first 15 years (1989–2003) using the longest available Ku-band VOD following the approach of ref. 41 (Methods and Supplementary Fig. 13), the underpinning interpretations need further validations. Nonetheless, independent analyses from VOD imply that water-sensitive semi-arid ecosystems might have become more important for CGR IAV during the past three decades. Variations in other constituent carbon fluxes are still not available; for instance, the temporal dynamics of large-scale soil respiration and the temperature sensitivity of soil respiration are constrained by data availability and thus remain uncertain[43]. Other possible mechanisms that are not investigated here include synergy effects of other drought-induced disturbances (such as fire effects[44]), lasting effects of tree mortality on carbon uptake[45] and decreasing temperature sensitivity of both tropical photosynthesis and/or soil respiration under global warming[46]. The most recent 30 years (1989–2018) overlaps with the decadal 'global warming hiatus' (1998–2012) in which natural internal variability, like ENSO, might play a part[47]. The effects of internal variability should not change the increased tropical water control on CGR IAV in the future because we have already taken ENSO into account. Further studies are required to investigate these potential mechanisms, for instance by integrating new observations from flux towers, field experiments and satellites[48] to calibrate process-oriented models.

In summary, we demonstrate that tropical water availability is likely to have increasingly controlled interannual atmospheric CGR over the past 59 years. The dominant climate driver of the interannual variation of terrestrial carbon cycle has already shown a tendency to shift from temperature to water, suggesting rising water limitations on tropical terrestrial carbon sink. We therefore also partly reconcile the debate of water versus temperature controls for the land carbon cycle[7,9,11] from the perspective of the considered time frames, in addition to the recent view that land–atmosphere feedback matters[20]. As the sensitivity of terrestrial carbon uptake to temperature is usually used as the metric to diagnose or constrain the terrestrial carbon–climate feedbacks[49], we call for more attention to the relevance of tropical water in predicting next-year atmospheric CGR and we suggest it is timely to introduce water-based constraints on future tropical terrestrial carbon–climate feedbacks. Uncertainties in terrestrial carbon–climate feedbacks strongly affect the assessment of the magnitude of emission reductions required to achieve any global temperature target. Hence, the failure of state-of-the-art models to capture the observed increasingly negative coupling between tropical land water and interannual CGR calls for a better characterization of relevant processes to improve the representation of the terrestrial carbon cycle in ESMs and climate projections.

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

# Article

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

# Methods

## CGR

Annual global atmospheric CGR spanning from 1960 to 2018 is obtained from the Greenhouse Gas Marine Boundary Layer Reference of the National Oceanic and Atmospheric Administration (NOAA/ESRL)[50]. According to the guideline, the annual CGR in a given year is the difference in $CO_2$ concentration between the end of December and the start of January of that year. In addition, we also use the estimated RLS from the latest Global Carbon Budget 2020[1] to verify the robustness of our main finding (Extended Data Fig. 1b). RLS is inferred as a residual between emissions, atmospheric $CO_2$ accumulation and the ocean sink.

## Climate data

**Terrestrial water storage and lagged precipitation.** The twin GRACE satellites provide the measurement of changes in terrestrial WS at monthly scale since March 2002[51]. In particular, terrestrial WS is the sum of all above- and below-surface WS, including soil moisture, groundwater, snow, ice and water stored in vegetation and rivers and lakes. To complement the shorter record of observations provided by the GRACE satellites, we use a recently published statistical reconstruction of terrestrial WS that trained on the twin GRACE satellites[18]. The reconstructed terrestrial WS is based on two different GRACE solutions and three different meteorological forcing datasets. Here, we mainly use the ensemble mean of all members. The details of the statistical approach are documented in ref. 18. Validation against original terrestrial WS supports the reliability of reconstructions in reproducing historical signals at yearly time scale, including tropics (Supplementary Fig. 14 and Supplementary Table 1). The reconstructed terrestrial WS dataset has a spatial resolution of $0.5° × 0.5°$ and a temporal resolution of one day from 1901 to 2018.

In addition, direct observations of LagP over tropical land are found to capture IAV of aggregated tropical WS anomaly at yearly time scale (Extended Data Fig. 3a and Supplementary Table 1). For instance, the LagP in 2018 is the sum up of the precipitation from July in 2017 to June in 2018. Moreover, consistent with previous findings[7,17], the IAV of tropical land lagged precipitation correlates with CGR well (Extended Data Fig. 3b). This helps to explain the rationality of this relationship from the process of WS memory. Therefore, LagP is identified as another efficient proxy for aggregated tropical WS anomaly IAV. Precipitation is obtained from station-based Climate Research Unit (CRU) TS4.03 (ref. 52) and Global Precipitation Climatology Centre (GPCC) Full Data Monthly v.2020. To further confirm that the data quality of station-based tropical precipitation is reliable, we compare station-based CRU precipitation with satellite-based Tropical Rainfall Measuring Mission (TRMM) precipitation (Extended Data Fig. 7a); we also compare CRU precipitation with GPCC precipitation which has much larger gauge stations than CRU (Extended Data Fig. 7b). In fact, compared to the 2000s to 2010s (in which satellite observations confirm the reliability of station-based tropical precipitation IAV), the number of gauge stations was much larger in the 1960s–1990s (Extended Data Fig. 7c). These validations suggest that station-based tropical precipitation IAV is reliable during 1960–2018. TRMM 3B43 precipitation dataset has a spatial resolution of $0.25° × 0.25°$ and a temporal resolution of 1 month from 1998 to 2019. Both CRU and GPCC have a spatial resolution of $0.5° × 0.5°$ and a temporal resolution of 1 month from 1891 to 2018.

Therefore, we use yearly reconstructed tropical WS and LagP to indicate the IAV of tropical terrestrial water availability anomaly.

**Temperature.** Temperature is obtained from CRU TS4.03 (ref. 52). The Berkeley Earth global surface temperature is also used for a robust test[53]. The two temperature datasets both have a spatial resolution of $0.5° × 0.5°$ and a temporal resolution of 1 month from 1901 to 2018.

**Regional domain definition.** Tropical lands are defined as the spatial average over all the vegetated land areas between 24° N and 24° S, as

in ref. 7. Tropical semi-arid region domain consists of shrubland and (woody) savannah, which is identified according to the land-cover classification map from MODIS (MCD12C1, type3). The map was regridded using a majority filter to a spatial resolution of $0.5° × 0.5°$.

**ENSO indices.** The ENSO is Earth's most important source of interannual climate variability. The areal averages of sea surface temperature (SST) anomalies relative to a long-term average climatology are used to characterize ENSO. The time period of 1960–2018 is used as the climatology here. The SST anomalies over Nino3.4 region (5° N–5° S,120° W–170° W) are most commonly used and included the signals of both EP ENSO and CP ENSO[54]. Year is considered EP ENSO when the largest DJF (December to February) SST anomaly over the region of 2° S–2° N, 110° E–90° W lies in the EP (east of 150° W) and Nino3 index exceeds 1 s.d. Year is considered CP ENSO when the corresponding largest DJF SST anomaly lies in the CP (west of 150° W) and Nino4 index exceeds 1 s.d. We note that the identification of ENSO types could vary with the method used[27]. This study uses the Extended Reconstruction Sea Surface Temperature (ERASST v.5) from the National Oceanic and Atmospheric Administration (NOAA). This dataset has a temporal resolution of 1 month from 1855 to the present and a spatial resolution of $2° × 2°$ (ref. 55).

## CMIP6 models

**Coupled ESM.** Nine coupled ESMs participating in the sixth phase of Coupled Model Intercomparison Project (CMIP6) are used: CESM2, CNRM-ESM2-1, IPSL-CM6A-LR, MPI-ESM1-2-LR, UKESM1-0-LL, ACCESS-ESM1-5, CanESM5, MIROC-ES2L and NorESM2-LM. Coupled ESMs allow for feedbacks between the physical climate and the biological and chemical processes in the ocean and on land. We adopted data output from the 'historical' scenario (1960–2014) with one ensemble member for each model. Climate and land carbon sinks are simulated with all-forcings, including both the natural causes (for example, volcanic eruptions and solar variability) and human factors (for example, $CO_2$ concentration, aerosols and land use) over the period 1850–2014. In coupled ESMs, the carbon cycle is coupled to the climate system.

**Offline LSM.** Six offline LSMs from the Land Surface, Snow and Soil Moisture Model Intercomparison Project (LS3MIP) are used here[56]: CESM2, CNRM-ESM2-1, IPSL-CM6A-LR, MPI-ESM1-2-LR, UKESM1-0-LL and CMCC-ESM2. The offline LSMs account for land-use changes but do not include local land–atmosphere feedbacks. We adopted data output from the 'Land-Hist' scenario (1960–2014) with one ensemble member for each model, for which the atmospheric forcing, vegetation, soil, topography and land/sea mask data were prescribed following the protocol used for the CMIP6 DECK simulations. The atmospheric forcing comes from the Global Soil Wetness Project phase three (GSWP3), which is a dynamically downscaled and bias-corrected version of the Twentieth Century Reanalysis[57]. Spin-up of the land-only simulations follow the protocol of the project 'Trends and drivers of the regional scale sources and sinks of carbon dioxide' (TRENDY)[58].

Following previous efforts[11,59], to enable a fair comparison of the water–carbon relationship between observations and models, we use the sum of soil moisture in all layers and snow water equivalent as modelled terrestrial WS. In the tropics, snow water equivalent is negligible.

**Partial correlation.** Partial correlation is used here to directly check whether increasingly negative water–carbon coupling is influenced by confounding water–temperature coupling. However, using specific values of $R_{W,CGR|T}$ to conclude the sign and strength of total water impacts on CGR is not suggested. $R_{W,CGR|T}$ isolates water impacts on CGR from confounding water–temperature coupling by linearly removing all temperature-related covariations. However, given the well-documented soil moisture–atmosphere feedbacks[21], temperature

# Article

variability actually includes many feedbacks from soil moisture (for example, hot extremes at tropical semi-arid regions) and removing all of them would indirectly remove some water impacts on CGR because of their physical connection. In addition, models do not reproduce the intensified water–carbon coupling but roughly capture the sign and strength of long-term tropical water–carbon coupling during 1960–2018, thus providing insights into underpinning processes. Model factorial experiments show that removing soil moisture interannual variability suppresses land carbon uptake variability by about 90%, whereas tropical mean temperature remains unchanged (Extended Data Fig. 10 in ref. 20). Therefore, ref. 20 suggest that tropical mean temperature might not represent a mechanistic climatic driver for land carbon uptake variability. Hence, $R_{W,CGR|T}$ is an insufficient and less accurate measure to infer the sign and strength of independent water impacts on CGR and underestimate water impacts on CGR in phases in which temperature control is dominant. Nonetheless, their relative changes are useful in this study and support the finding that water–carbon coupling has become increasingly negative in the recent past (1989–2018) compared to previous climate conditions (1960–1989).

**Empirical orthogonal function.** The method of empirical orthogonal function (EOF) analysis can deconvolve the spatiotemporal variability of a signal into orthogonal modes, each indicated by a principal spatial pattern and the corresponding principal component time series. It is widely used to study spatial patterns of climate variability and how they change with time[60,61]. We perform the EOF analysis on simulated tropical soil moisture from CMIP6 models.

**Ridge regression.** In the presence of collinearity, using the OLS estimator can lead to regression coefficient estimates that have large sampling variability and even a wrong sign. Ridge regression is a common technique to be used to address the issues arising through collinearity[62]. In ridge regression, a penalty term is added to the loss function to shrink the regression coefficients[63]. The amount of shrinkage is defined by a regularization parameter that was chosen by us in a cross-validation approach. The data were randomly split 25 times into training and validation sets and, for each split, the performance for the validation test set (mean squared error) was assessed for 100 different regularization parameters spaced evenly on a log scale. The best-performing regularization parameters were selected for each split and the average between them was retained for the final model. To assess the uncertainty in the regression coefficient estimates, we relied on bootstrapping, meaning that we randomly sampled the data 5,000 times and estimated the regression coefficients for each sample.

**Spatial coherence.** To quantify the degree of spatial coherence of yearly tropical WS anomaly, following ref. 9, we calculate a large covariance matrix of all grid cells versus all grid cells for tropical WS anomaly. Each element in this covariance matrix is termed as $c_{i,j}$ as follows:

$$c_{i,j} = \mathrm{cov}(WS_i, WS_j) \tag{1}$$

where $i$ and $j$ indicate the two grid cells that used to calculate covariance, $WS_i$ and $WS_j$ are the corresponding yearly WS anomaly time series in the specified time period. Then, we summed all, positive and negative covariance terms (termed as tcov, tcov$^+$ and tcov$^-$), respectively, as follows:

$$
\begin{aligned}
\mathrm{tcov} &= \sum_{i=1}\sum_{j\neq i} |c_{i,j}| \\
\mathrm{tcov}^+ &= \sum_{i=1}\sum_{j\neq i} c_{i,j}|c_{i,j} > 0 \\
\mathrm{tcov}^- &= \sum_{i=1}\sum_{j\neq i} c_{i,j}|c_{i,j} < 0
\end{aligned}
\tag{2}
$$

The variances in the diagonal of the covariance matrix (where $i = j$) were excluded because they are always positive and do not contribute to the estimate of spatial coherence. Finally, the spatial coherence of WS anomaly was defined as the following equation:

$$\text{Spatial coherence} = \frac{\mathrm{tcov}^+ + \mathrm{tcov}^-}{\mathrm{tcov}} \times 100\% \tag{3}$$

In theory, 100% indicates all grid cells covariate in the same sign, that is, highest spatial coherence. Lower values indicate that total positive covariances are counterbalanced by total negative covariances, that is, lower spatial coherence.

**Vegetation optical depth and aboveground carbon.** VOD retrieved from microwave satellite observations is linked to the water content of vegetation mass and offers possibilities for monitoring AGC dynamics[41]. We used recently published long-term VOD products from the VOD Climate Archive (VODCA), which combines VOD retrievals that have been derived from multiple sensors (SSM/I, TMI, AMSR-E, WindSat and AMSR2) using the Land Parameter Retrieval Model[64]. For time completeness, we used the longest available VOD estimated from Ku-band which covers the period 1988–2016. To estimate the tropical AGC, following the approach of ref. 65, we first fitted a four-parameter empirical function by calibrating tropical VOD against the tropical AGC benchmark map from ref. 66 in 2000 as follows:

$$\mathrm{AGC} = a \times \frac{\arctan(b \times (\mathrm{VOD} - c)) - \arctan(-b \times c)}{\arctan(b \times (\mathrm{Inf} - c)) - \arctan(-b \times c)} + d \tag{4}$$

where $a$, $b$, $c$ and $d$ are four best-fit parameters and Inf is set to $10^{10}$. AGC density (MgC ha$^{-1}$) was derived by multiplying the original aboveground biomass density values by a factor of 0.5 (ref. 67). For Ku-VOD and AGC, data are aggregated to the spatial resolution of $0.5° \times 0.5°$. The spatial scatter plot of VOD and AGC clearly demonstrates the good relationship between VOD and AGC (coefficient of determination $R^2 = 0.76$, $P < 0.01$; Supplementary Fig. 15). It seems the performance of VODCA Ku-VOD is less comparable to L-VOD ($R^2 = 0.81$) (ref. 41), which was considered to be more sensitive to AGC in high biomass regions. However, L-VOD has only been available since 2010 and thus is not used here. Finally, we apply this empirical function to convert VOD to AGC from 1988 to 2016.

## Data availability

All the datasets used here are publicly available. Atmospheric $CO_2$ observations are available at https://gml.noaa.gov/ccgg/; GRACE observations of terrestrial WS are available at https://grace.jpl.nasa.gov/data/get-data/monthly-mass-grids-land/; GRACE-REC terrestrial WS are available at https://doi.org/10.6084/m9.figshare.7670849; CRU climate datasets are available at https://www.uea.ac.uk/groups-and-centres/climatic-research-unit; GPCC precipitation dataset is available at https://www.dwd.de/EN/ourservices/gpcc/gpcc.html; TRMM precipitation dataset is available at https://disc.gsfc.nasa.gov/datasets/; Berkeley Earth climate datasets are available at http://berkeleyearth.org/; ERASST v.5 are available at https://psl.noaa.gov/data/gridded/data.noaa.ersst.v5.html; CMIP6 model outputs are available at https://pcmdi.llnl.gov/CMIP6/; and VODCA products are available at https://doi.org/10.5281/zenodo.2575599.

## Code availability

Codes are available through Zenodo at https://doi.org/10.5281/zenodo.6447779.

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

**Acknowledgements** We acknowledge the World Climate Research Programme, which, through its Working Group on Coupled Modelling, coordinated and promoted CMIP6. We thank the climate modelling groups for producing and making available their model output, the Earth System Grid Federation (ESGF) for archiving the data and providing access and the multiple funding agencies who support CMIP6 and ESGF. We thank all contributors to the LS3MIP and LMIP experiments. We thank the Global Monitoring Division of NOAA/Earth System Research Laboratory for providing the atmospheric $CO_2$ measurements. We thank U. Beyerle, L. Brunner, R. Lorenz and M. Hauser for downloading and processing the CMIP6 and LS3MIP data. We acknowledge the NOAA/OAR.ESRL PSL for providing the ERSST_v5 dataset. We thank M. Hirschi for downloading the VOD data. We acknowledge V. Humphrey and Q. Sun for useful discussions. L.L., R.S.P., P.C., P.F. and S.I.S. acknowledge support from the European Union's Horizon 2020 Research and Innovation Programme (grant no. 821003 (4C)). L.L., P.C. and S.I.S. also acknowledge support from HORIZON.2.5 (grant no. 101056939) (RESCUE).

**Author contributions** L.L. conceived the original idea. L.L., S.I.S. and P.C. designed the experiments. L.L., S.I.S., P.C., W.M., R.S.P., P.F., J.S. and L.G. performed the research. J.S. carried out ridge regression analyses. L.L. carried out all other analyses. L.L. wrote the paper with contributions from all co-authors.

**Funding** Open access funding provided by Swiss Federal Institute of Technology Zurich.

**Competing interests** The authors declare no competing interests.

**Additional information**
**Correspondence and requests for materials** should be addressed to Laibao Liu.

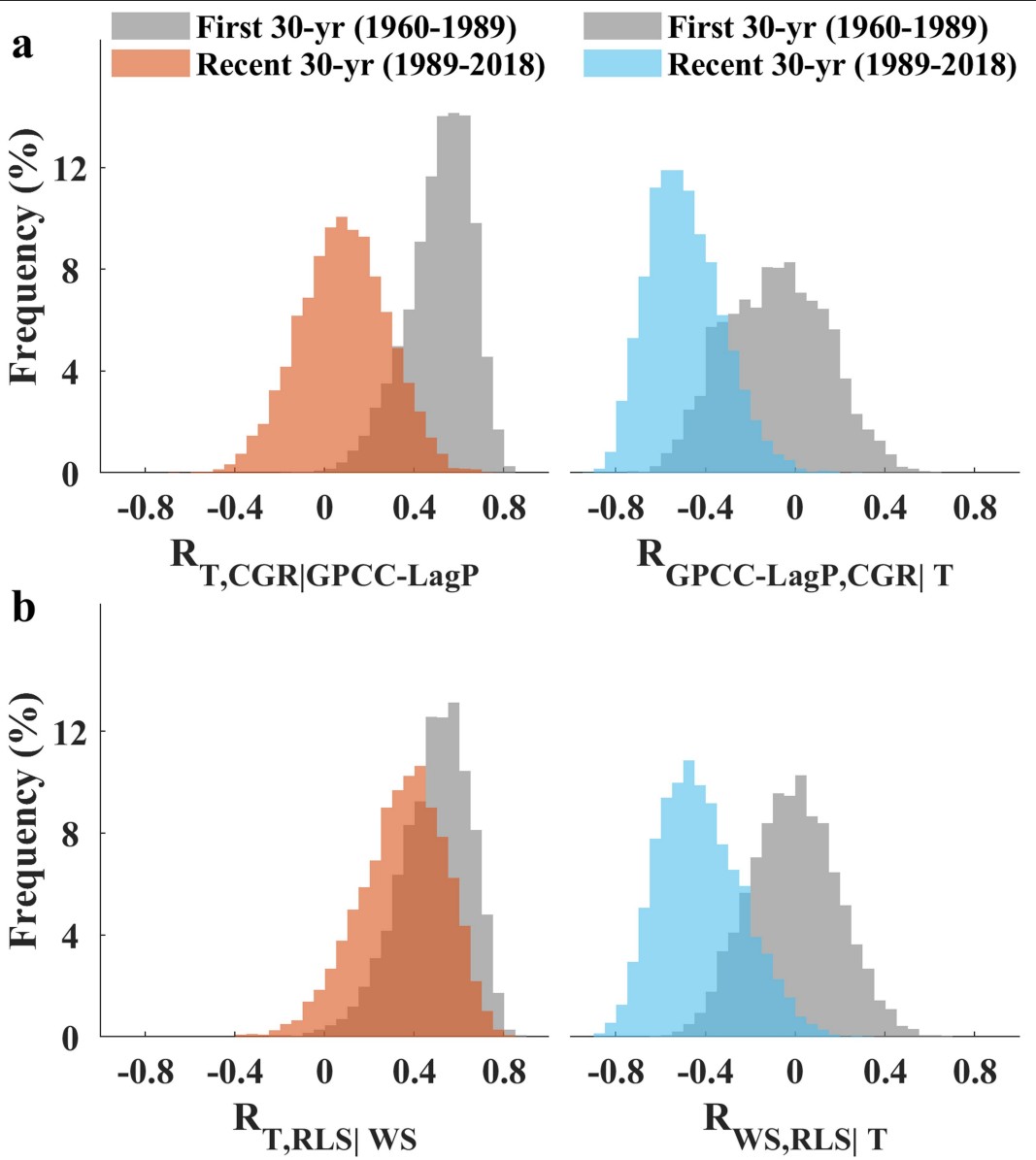

**Extended Data Fig. 1 | Robustness of tropical land climate–carbon interannual correlations. a**, Histograms of climate–carbon interannual correlations in the first three decades (1960–1989) and in the recent three decades (1989–2018), derived using 5000 bootstrapping repeats. Same as Fig. 1c, but tropical water refers to 6-month lagged precipitation from GPCC and tropical temperature is derived from Berkeley Earth global surface temperature. **b**, Same as Fig. 1c, but CGR is replaced with residual land sink (RLS).

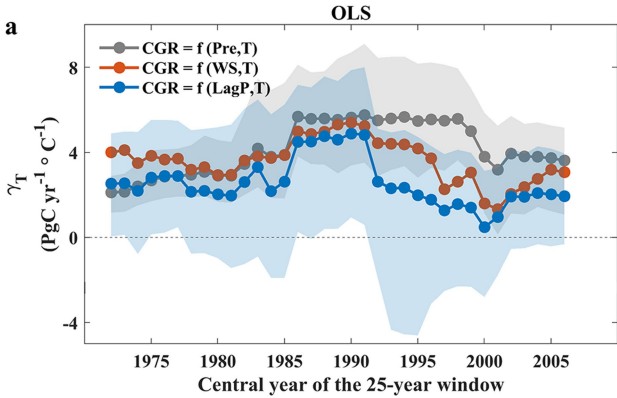

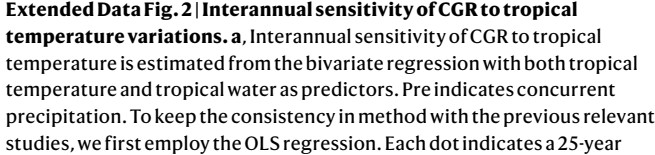

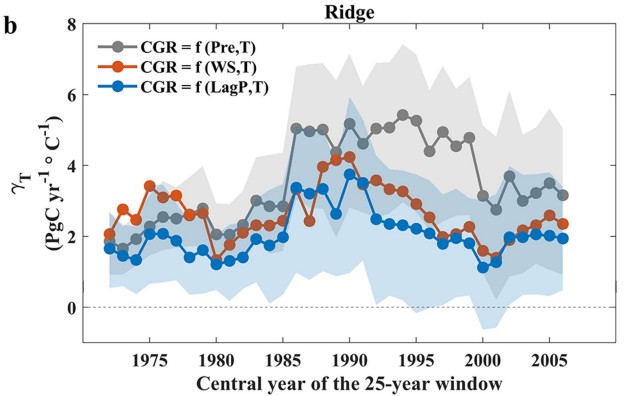

**Extended Data Fig. 2 | Interannual sensitivity of CGR to tropical temperature variations. a**, Interannual sensitivity of CGR to tropical temperature is estimated from the bivariate regression with both tropical temperature and tropical water as predictors. Pre indicates concurrent precipitation. To keep the consistency in method with the previous relevant studies, we first employ the OLS regression. Each dot indicates a 25-year period. The central year of the time window is labelled on the horizontal axis. Shaded areas represent the 95% confidence interval, derived using 5000 bootstrapping repeats. For a better readability, only the 95% confidence interval of temperature sensitivity from the first and third regression function are plotted. **b**, same as a, but using the Ridge regression.

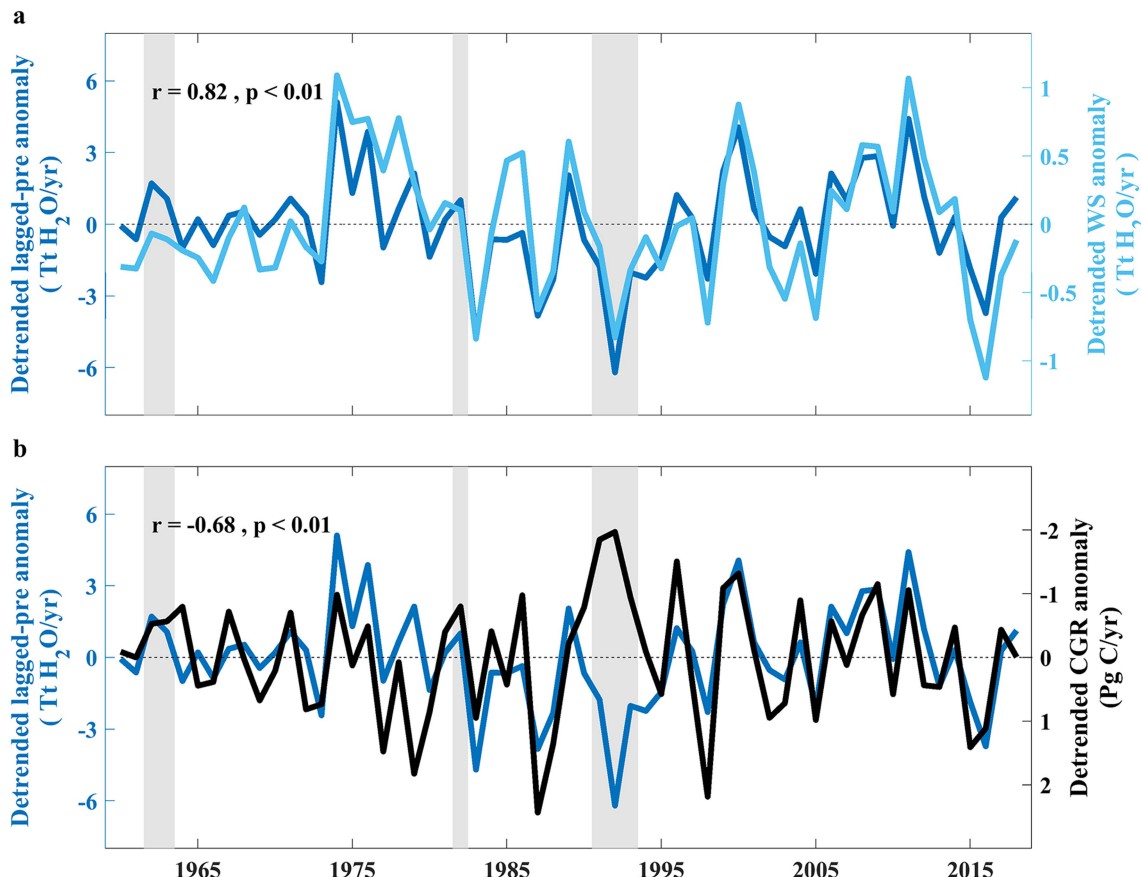

**Extended Data Fig. 3 | Suitability of using LagP as the poxy of tropical terrestrial water IAV. a, b,** IAV of tropical LagP and (a) tropical WS and (b) CGR during 1960–2018. Vertical grey shades indicate three volcanic eruptions (Mount Agung, El Chichón and Pinatubo).

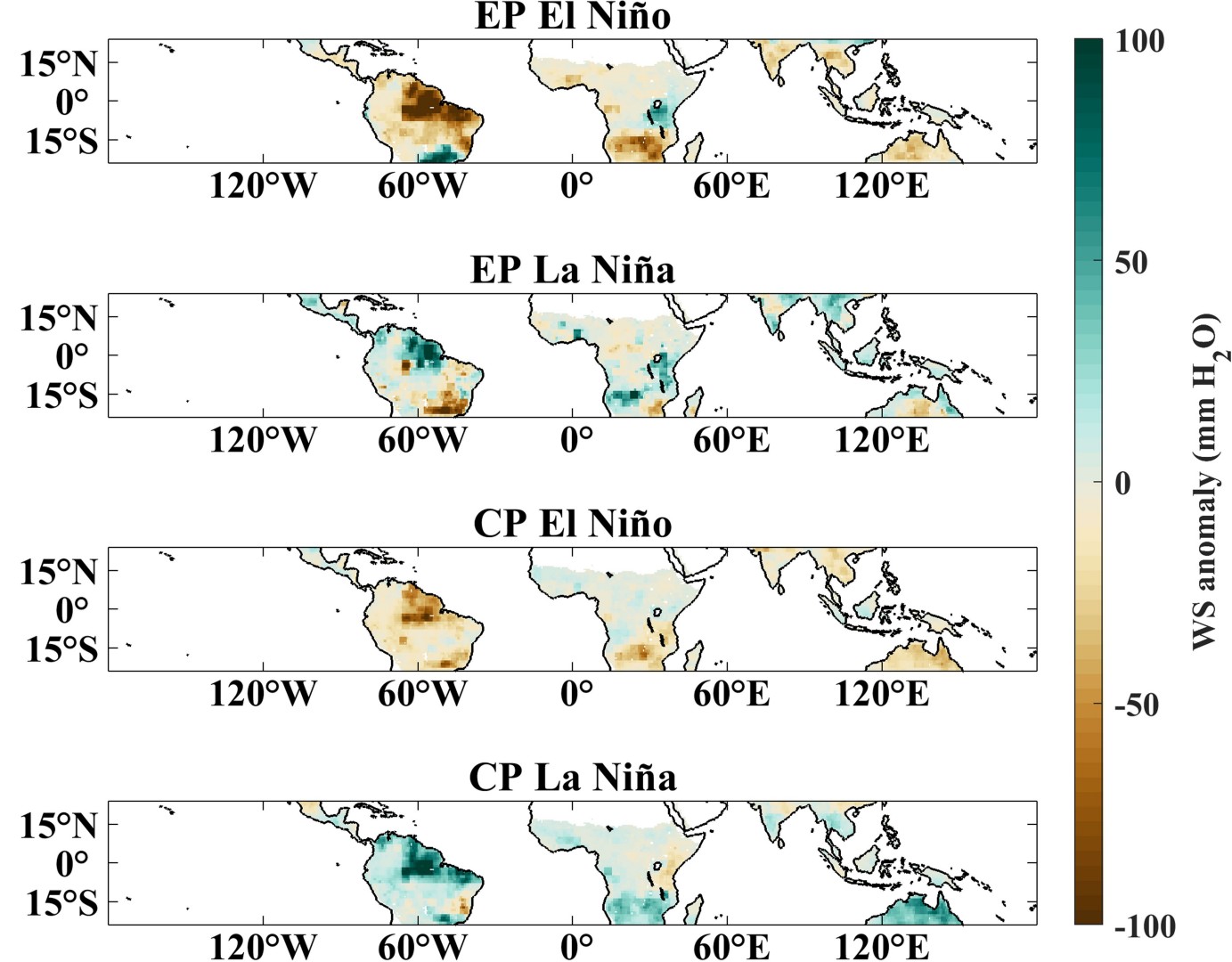

**Extended Data Fig. 4 | Spatial pattern of WS anomaly at EP ENSO and CP ENSO years.** Year is considered EP ENSO when the largest DJF SST anomaly over the region of 2°S–2°N, 110°E–90°W lies in the Eastern Pacific (Eastern of 150°W) and Nino3 index exceeds one standard deviation. Year is considered CP ENSO when the corresponding largest DJF SST anomaly lies in the Central Pacific (Western of 150°W) and Nino4 index exceeds one standard deviation. Volcano years are excluded from analyses.

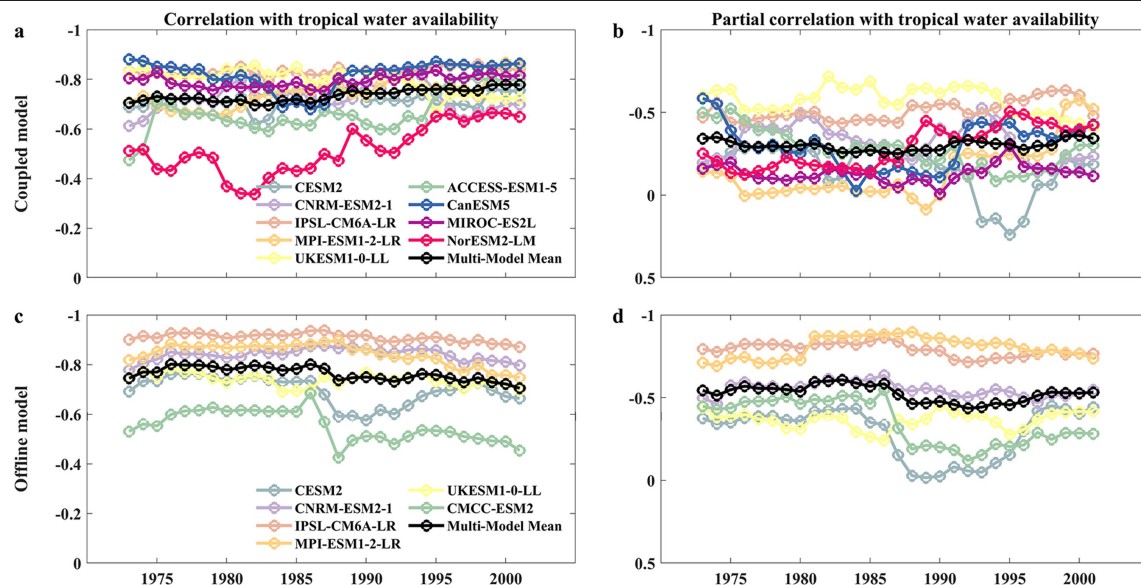

**Extended Data Fig. 5 | Interannual correlation of land carbon fluxes to tropical water in models.** Years labelled on the horizontal axis indicate the central year of the 27-year moving time window (all variables detrended at yearly scale in each corresponding window). Models are based on tropical total soil moisture, tropical temperature and global net ecosystem exchange.

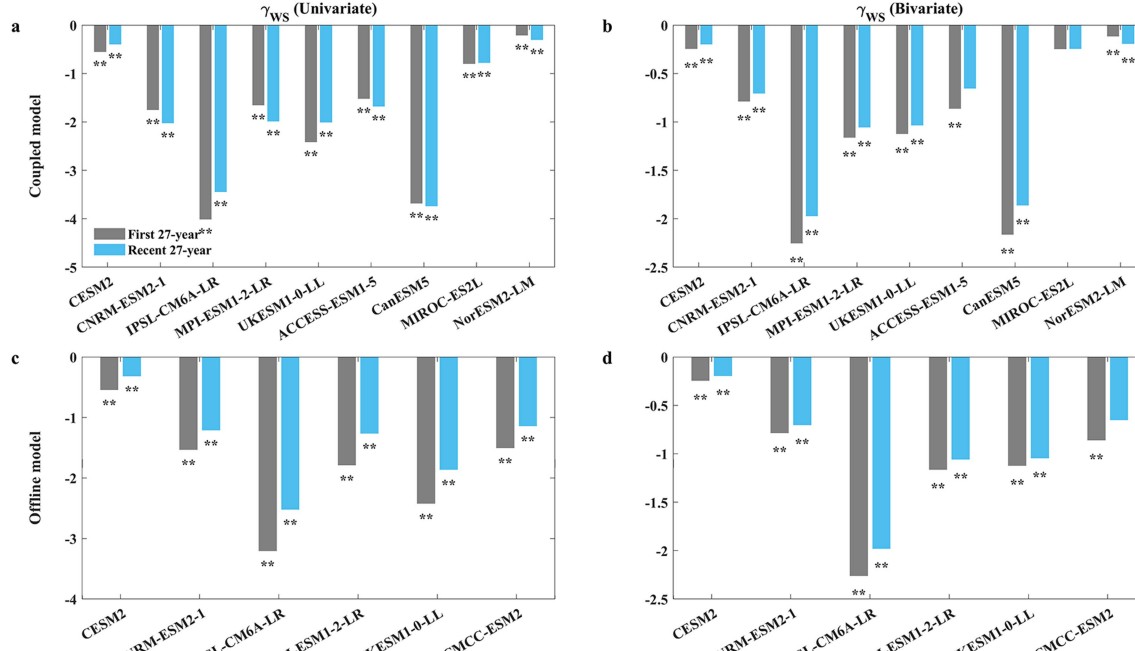

**Extended Data Fig. 6 | Interannual sensitivity of land carbon fluxes to tropical water in models.** For models, global net ecosystem exchange, tropical total soil moisture, and tropical temperature from each model are used for computations. Univariate and bivariate sensitivity are estimated using the OLS regression and Ridge regression, respectively. The best estimate of sensitivity of CGR to tropical water is shown. ** indicates a significant sensitivity at P < 0.05.

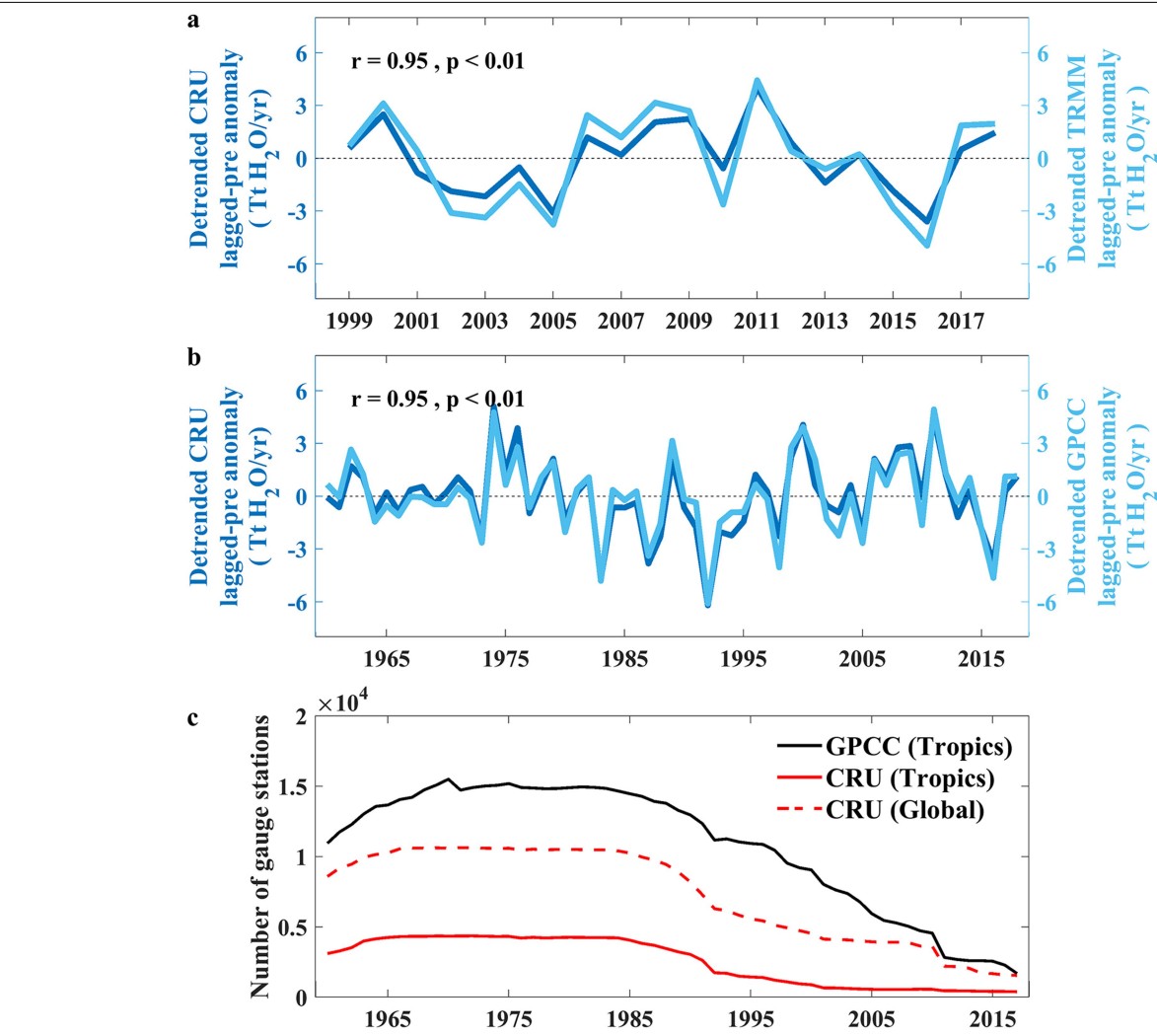

**Extended Data Fig. 7 | Robustness of tropical land LagP IAV. a**, **b**, IAV of tropical LagP in CRU, (a) TRMM during 1999–2018 and (b) GPCC during 1960–2018. **c**, Temporal coverage of the number of gauge stations in GPCC and CRU.

**Extended Data Table 1 | Interannual correlation and sensitivity of CGR to tropical water**

| Metric | Method | Proxy of Water | Metrics during first 30-year | Confidence level for statistical significance of Metrics | Metrics during recent 30-year | Confidence level for statistical significance of Metrics | Test whether the Metrics become more negative (P value and corresponding confidence level) |
|---|---|---|---|---|---|---|---|
| **Correlation** | R(W,CGR) | WS | $-0.53 \pm 0.14$** | 99.9% | $-0.71 \pm 0.10$** | 99.9% | P = 0.10 (89.6%) |
| | | LagP | $-0.58 \pm 0.15$** | 99.2% | $-0.80 \pm 0.06$** | 99.9% | P = 0.03 (97.2%) |
| | R(W,CGR\|T) | WS | $0.13 \pm 0.21$ | 44.5% | $-0.36 \pm 0.20$* | 90.4% | P = 0.08 (91.8%) |
| | | LagP | $-0.04 \pm 0.24$ | 2.7% | $-0.56 \pm 0.14$** | 99.6% | P = 0.03 (97.2%) |
| **Sensitivity** | Univariate (OLS) | WS | $-0.95 \pm 0.27$** | 99.9% | $-1.26 \pm 0.23$** | 99.9% | P = 0.11 (88.6%) |
| | | LagP | $-0.24 \pm 0.07$** | 99.4% | $-0.33 \pm 0.05$** | 99.9% | P = 0.12 (87.7%) |
| | Bivariate (Ridge) | WS | $-0.05 \pm 0.33$ | 15.6% | $-0.66 \pm 0.26$** | 98.9% | P = 0.09 (90.6%) |
| | | LagP | $-0.06 \pm 0.10$ | 43.4% | $-0.22 \pm 0.05$** | 99.9% | P = 0.09 (90.7%) |
| | Bivariate (OLS) | WS | $0.33 \pm 0.55$ | 43.4% | $-0.68 \pm 0.44$* | 90.4% | P = 0.09 (90.6%) |
| | | LagP | $-0.03 \pm 0.13$ | 4.9% | $-0.24 \pm 0.07$** | 99.7% | P = 0.09 (90.7%) |
| | | | | 55.3% (On average) | | 97.9% (On average) | 91.5% (On average) |

Estimates are derived from 5000 bootstrapping repeats by randomly selecting years without volcano perturbations in each sub-period. The mean, one standard deviation, and the corresponding confidence level for statistical significance are presented for the metric. For the statistical test in the last column, we first generate bootstrapping repeats in each sub-period and compute the difference in each metric between sub-periods. To derive fully independent samples of data for two sub-periods, the overlapping year 1989 is excluded. The bootstrap distribution of the differences is then used to estimate the probability of observing an effect as or more negative than what was actually observed if there was really no difference (i.e., the single-sided P value). We note that, because the annual CGR records are short (n < 30 per group), the uncertainty of correlations and regression coefficients is inherently large, and the statistical power of tests for between-group differences is limited. Consequently, relatively large P values (e.g., P=0.1) should be interpreted with caution, as they carry a higher probability of failing to detect a real change. Accordingly, the magnitude of correlations and sensitivities should play an important role when interpreting the results, rather than the P value alone. ** and * indicate a significant sensitivity or correlation at P < 0.05 and P < 0.1, respectively.