## [Peer Review File · Nature]

Manuscript Title: Increasingly negative tropical water - interannual CO₂ growth rate coupling

Reviewer Comments & Author Rebuttals

Reviewer Reports on the Initial Version:

Referees' comments:

Referee #1 (Remarks to the Author):

Liu et al. explore the role of the tropics in controlling the inter-annual variability (IAV) in the atmospheric growth rate of CO₂, in particular, they focus on two time periods to show the reln. has intensified/strengthened. They speculate as to why this might be the case and finally, demonstrate that this behaviour isn't captured by models, but broadly supported by remote sensing (albeit of different variables and much bigger changes).

Overall, I think this is an interesting analysis, one I would certainly want to read after some careful revisions. I think it really needs to make much better use of the modelling data. I don't learn anything by saying "obs and models are different" (see a suggestion below). There is a lot of discussion in the manuscript about IAV, but really isn't this paper not actually about IAV but instead the trend? Perhaps the authors disagree, but I'm having a hard time reconciling these things. Finally, I'd urge the authors to think about what the big take-home is: are we saying that the tropical system is interacting with the climate in a functionally new way for the last 30 yrs? How can we be sure 60 years is sufficient to draw this conclusion (I'm sure you could argue because the climate is relatively stable vs other systems, for example...). The authors seem to posit this is due to CP vs EP, is this the main explanation, what else? And how much of the variability in terrestrial carbon fluxes do these ENSO teleconnections explain? Finally and this may be difficult to address so ignore as you wish ... this study focussed on the tropics, I guess the context I lack is whether this behaviour shown is seen globally (but "we've" only looked at the tropics), or whether this is only seen in the tropics, in which case this is a hugely important story and unique to the tropis...

Other thoughts...

Perhaps there isn't the space, but I found myself wanting to first be told how much tropical temperatures vary. We begin by being told there is a correlation between CGR and tropical temperatures. But one naturally assumes tropical temperatures are (relative) less variable than other locations. Would it be helpful to add a sentence with this as the starting point before you discuss correlation?

While Figure 1a is very interesting, I find myself wondering about what the temperature (water) anomaly means in the context of ecosystem behaviour. To explain, the temperature anomalies are relatively small and represent an annual average. Mechanistically, how much difference would we expect to predict in carbon uptake/respiration for these magnitudes of anomaly? I dare say that we wouldn't actually hypothesise we'd be easily able to separate variations between years, owing to the

variability in the timing of how these anomalies would present themselves seasonally as a forcing. I guess, seeing as you have the models, would these patterns look like? I would imagine far, far messier, which one could attribute to poor model skill, but perhaps in reality, also that this reln should be more variable...? While ~18 (roughly by eye) of the biggest CGR anomalies are in the hot & dry, ~9 are not, including two of the biggest anomalies. So, I guess, yes there is a pattern and also, it isn't as simple as the graph is trying to suggest? I don't think the text really reflects what you see by looking at the plot, instead it opts for the simpler summary without pointing to the examples that don't fit the pattern. I think this matters and perhaps strengthens the argument: 1a is showing to me there is still marked IAV and it isn't easily attributable to just been hot & dry or the opposite. By contrast, the rest of figure 1 is nicely showing that despite this variability, the trend is clear. My general take from this figure is that the variability in CGR isn't necessarily well understood on IAV timescales but we can still see that the role of water availability is driving a background change. I feel like this isn't necessarily the interpretation the authors have though...or at least, I'm not getting this from the text (it may just be me!).

I really like the analysis in 1b-e (which was nicely supported by an alternative approach in the supplement).

Lines 43, 47, 50: It would be good to tell the reader how strong the reln between CGR and ENSO is - what is the correlation? Similarly, the correlation for lagged precipitation and CGR? And GRACE and CGR. In each instance the reader is told it is "strong"... These statements appear liberally throughout the text. For example, Zhang et al (<https://onlinelibrary.wiley.com/doi/full/10.1111/gcb.14275>) say ENSO is explaining <30% of productivity (I think from memory) - so with this in mind - is it worth some discussion about other drivers beyond ENSO? It may be that this study isn't breaking this down into EP vs CP, I honestly can't recall - but my general point is with a little editing, the authors could better inform the reader...and I think it would enhance the manuscript.

With the coupled model analysis - the text isn't clear, during the historical periods, is each model following the same climate forcing patterns, in CMIP6 (presumably not, i.e. for each historical period, each model has its own random sequence of meteorology)? I doubt this will be obvious - can this be explained to the reader in the methods? As such, I cannot possibly see how this comparison makes sense? Forcing the LSMs with the same climate as you see during the period makes more sense, but probing the CMIP models?

I understand that with the multi-models, it is hard to recreate the analysis shown for the obs, but I do feel it would at least be helpful to show each model as a version of Fig 1? I would really like to see a version for NEE for the histograms (even if in the supplement). Fundamentally, I find the model comparison extremely weak. It just says things are different and we need improvements. This actually entirely unhelpful. How are these results going to motivate the models to improve? Can we at all diagnose something more useful? As I suggested above - what is our expectation re: the variability to these anomalies? For example, the authors hypothesise that a driving factor in the obs is the change in the teleconnections, shift from EP to CP - do we think the obs forcing captures this in the offline met driving? Are the models not capturing the change in coupling because it isn't in the driving data or because the processes are wrong? Given you have also shown the coupled models, then I think you can also answer about whether the coupled models are capable of simulating the

change in CPvsEP - presumably not... Is there one model or a few models that do simulate the CP vs EP change and do they better simulate the change?

As for the VOD - I'm have no expertise in these data, but I wonder if it makes sense the differences should be so pronounced!?? If we think VOD is a proxy for aboveground biomass, wouldn't we predict a smaller change in the stock vs the change in fluxes? Yet the different in histograms in S7 is far greater between periods! I presume this have accounted for land use change?

Finally, it shouldn't need to be said but: "Codes are available under reasonable requests" isn't acceptable. I urge the authors to put their code into a repository and make it freely available, regardless of what the journal's policy is.

Referee #2 (Remarks to the Author):

This paper investigates the temporal variability of the correlation between the carbon growth rate and both water availability and temperature variations and finds the former is strengthening. The paper further argues that ENSO changes and changes in spatial patterns of water storage (TWS) anomalies underly this pattern. Understanding how climate affects the carbon growth rate (CGR) through its effect on the terrestrial carbon cycle is a fundamental component of understanding climate change, so the paper's advances will be of interest to Nature's readership. The explanation the authors provide based on changes in the degree of spatial compensation is compelling. I also commend the authors for illustrating a key limitation in the Wang et al Nature 2014 analysis. However, I have several fundamental concerns about the robustness of the results, as well as the novelty of some of them.

1) The authors investigate the strength of the relationship between TWS and CGR and refer to this in many locations as the 'sensitivity' (e.g. lines 126, 128, etc). However, the sensitivity of the CGR-TWS relationship, and the quantity of interest for understanding the CGR, is the slope, not the correlation coefficient of this relationship. This quantity does not seem to be investigated anywhere.

2) The analysis relating water storage to ENSO type in Fig. 2d,e considers only the overall spatially averaged water anomaly, if I understand the analyses correctly. However, while this ENSO analysis is convincing, and the analysis in Figure 2a,b,c is convincing regarding the spatial coherence shift, it does not explicitly show that the change in spatial coherence of the water storage anomaly with time is directly attributable to the ENSO shift, as directly linked in lines 163-165 and implied elsewhere. Could these two factors not be independent?

3) The Humphrey et al, Nature, 2018 paper cited here already showed that DGVMs are unable to capture the CGR vs. water storage and CGR vs. temperature correlations. Since the process representations of DGVMs have much in common with uncoupled models, the failure of these models to capture the right trend in correlation when they can't get the overall average correlation right (as shown in Figure 3) is not really a big surprise/particularly novel. The discussion of missing processes also contains significant duplicity with that in Humphrey et al, 2018. This should be at least

acknowledged.

4) As the authors mention on line 241, VOD is sensitive to canopy water content. Thus, its interannual fluctuations (and thus correlations to interannual CGR changes), can be potentially driven by changes in biomass or by changes in relative water content. The analysis the authors show in Figure S10 only investigates the *spatial* correspondence between VOD and AGC, not the correspondence of temporal variations. It is therefore not sufficient to demonstrate the robustness of the lines 234-260. In fact, as shown in detail in Konings et al., GRL, 2021, VOD interannual fluctuations are likely a poor proxy for AGC interannual variability because of the strong influence of interannual variations in relative water content. Because relative water content changes are likely to be significantly affected by changes in water storage (and thus the water storage-CGR coupling investigated here), this analysis is unlikely to be robust and should be removed. Instead, why not simply look at GRACE and the CGR directly for either GRACE in semi-arid regions vs. GRACE in moist regions? This would be a more explicit, data-driven demonstration of the role of semi-arid regions. Although the coarse resolution of GRACE may add some noise, this analysis would likely be significantly more robust than the one currently in lines 234-260.

More minor comments:

* Lines 43-45: Use of the past passive voice here is confusing, as it is unclear whether you are talking about your own results or those of others.

* Line 86: The more recent periods (e.g. the 1994-2018 period mentioned in line 86 and others shortly before it), overlap with the decadal 'warming hiatus' period, which is at least partially caused by internal variability. See, e.g. Medhaug et al, 2017. How does this affect our expectations for whether this behavioral shift is likely to continue well into the future? At the very least some discussion on this general topic seems warranted.

* Figure 1b: It's a bit difficult to see that the blue axis are negative correlations. Perhaps this can be reflected in the axis label instead?

* What spatial and temporal space is Figure 2e calculated at?

References:

- Humphrey, V., Zscheischler, J., Ciais, P., Gudmundsson, L., Sitch, S., & Seneviratne, S. I. (2018). Sensitivity of atmospheric CO₂ growth rate to observed changes in terrestrial water storage. *Nature*, 560(7720), 628–631. <https://doi.org/10.1038/s41586-018-0424-4>
- Konings, A. G., Holtzman, N. M., Rao, K., Xu, L., & Saatchi, S. S. (2021). Interannual Variations of Vegetation Optical Depth are Due to Both Water Stress and Biomass Changes. *Geophysical Research Letters*, 48(16), e2021GL095267. <https://doi.org/10.1029/2021GL095267>
- Medhaug, I., Stolpe, M. B., Fischer, E. M., & Knutti, R. (2017). Reconciling controversies about the 'global warming hiatus.' *Nature*, 545(7652), 41–47. <https://doi.org/10.1038/nature22315>

Referee #3 (Remarks to the Author):

Liu et al. investigated the temporal trend of the relationships between tropical water and the interannual variability (IAV) of atmospheric CO₂ growth rate during 1960-2018, primarily using reconstructed total water storage data, precipitation from CRU, and the CO₂ growth rate from NOAA surface sites. They concluded that the tropical water availability is increasing controlling the IAV of the atmospheric CO₂ growth rate and thus the tropical terrestrial biosphere carbon cycle. The paper further analyzed CMIP models and several offline terrestrial biogeochemical models, and found that these models could not reproduce the intensified water - carbon coupling. As uncertainty in carbon-climate feedback is one of the leading causes in the uncertainties of climate projections, the topic of this study is important. However, I have a few major concerns. Please see below for my detailed comments.

1) The uncertainty for the linear fitting lines in Figure 1b and 1d do not seem to include the uncertainty in the original dataset. As total water storage (TWS) data is from modeling estimates, it is important to consider its uncertainties in the linear fitting. Figure S2 shows the sensitivities of CGR to climate variations, which do show a very large uncertainty.

2) The conclusion of increasing tropical water control on interannual CO₂ growth rate is not supported by the analysis. Controlling for temperature effect, the absolute value of $R(\text{WS}, \text{CGR} | T)$ has the same magnitude in late 1960s (0.4) as in early 2000s (-0.4) (Figure 1d), only that they have different sign. Thus, the tropical water storage has the similar magnitude of control on CGR in the late 1960s as in early 2000s when controlling temperature, but just in the opposite direction. TWS affects both plants growth and ecosystem respiration, and the net of which controls CGR. The effect of TWS on plants growth and respiration could have changed due to change of climate, which needs further investigation.

In Figure 2, the authors argued that the spatial coherence of WS anomaly is the reason for the changes in $R(\text{WS}, \text{CGR} | T)$. But since $R(\text{WS}, \text{CGR} | T)$ in late 1960s has the similar magnitude as that in 2000s, following the same argument, it also requires some coherence in WS anomaly.

3) The data quality of both reconstructed water storage (TWS) data and precipitation data from CRU are questionable over the tropics. Figure 5 in Humphrey and Gudmundsson (2019) show that the correlation between reconstructed TWOS and the original dataset is close to zero over most of the tropical Africa, and less than 0.3 over most of the tropical South America. Figure 6 in the same paper shows that the Nash-Sctcliffe efficiency is less than 0.2 (closer to 1 corresponds to better performance) over most of the tropical region. It is unclear to me why Figure S8 shows much higher correlation than in Humphrey and Gudmundsson.

Even though Figure S8 shows much higher correlation than in Humphrey and Gudmundsson, Rec-TWS does show poor performance of over tropical Africa, which is critical in the spatial pattern analysis in Figure 2a and b.

CRU is also a reconstructed dataset based on station data. Figure 11 in Harris et al. (2020) shows that correlation coefficient from cross-validation for precipitation is less than 0.3 over most of the tropical continents. In addition, the number of observation stations is very sparse over the tropics (Figure 1 in Harris et al., 2020). Coupled with the changing number of stations and the heterogeneity

of the precipitation in general, the variability analysis with the dataset is questionable. In Figure S1, there is a big jump in the $R(\text{pr}, \text{CGR})$ changing from close to 0 to 0.3 in 1990s. If it were not due to the change of data quality of change of observation stations, what could have caused that?

In Figure S1c and Figure 1d, there is also a jump in 1995.

Humphrey, V. and Gudmundsson, L.: GRACE-REC: a reconstruction of climate-driven water storage changes over the last century, *Earth Syst. Sci. Data*, 11, 1153–1170, <https://doi.org/10.5194/essd-11-1153-2019>, 2019.

Harris, I., Osborn, T.J., Jones, P. et al. Version 4 of the CRU TS monthly high-resolution gridded multivariate climate dataset. *Sci Data* 7, 109 (2020). <https://doi.org/10.1038/s41597-020-0453-3>

4) The proposed mechanism for the increasing carbon-water coupling and the analysis of terrestrial biosphere models (TBMs) is not coherent. The study proposed that the changes in ENSO effect on tropical TWS anomaly strengthen tropical water control on CGR. If that the case, the offline TBMs should be able to reproduce the increasing control of water on CGR, since these models used reanalysis as forcing.

Author Rebuttals to Initial Comments:

We thank the editor and the reviewers for their insightful comments and suggestions to the manuscript. The manuscript has been substantially revised following the reviewers' comments. **Most importantly, we highlight with new evidence that the results are robust, and that the expressed main concerns were due to misunderstandings, which we addressed with improved clarity of the text.**

Please see our detailed response below. The original comments are in black, and our responses are given in blue.

Reviewer' comments:

Reviewer #1 (Remarks to the Author):

Reviewer C1.0:

Liu et al. explore the role of the tropics in controlling the inter-annual variability (IAV) in the atmospheric growth rate of CO₂, in particular, they focus on two time periods to show the reln. has intensified/strengthened. They speculate as to why this might be the case and finally, demonstrate that this behaviour isn't captured by models, but broadly supported by remote sensing (albeit of different variables and much bigger changes).

Overall, I think this is an interesting analysis, one I would certainly want to read after some careful revisions. I think it really needs to make much better use of the modelling data. I don't learn anything by saying "obs and models are different" (see a suggestion below). There is a lot of discussion in the manuscript about IAV, but really isn't this paper not actually about IAV but instead the trend? Perhaps the authors disagree, but I'm having a hard time reconciling these things. Finally, I'd urge the authors to think about what the big take-home is: are we saying that the tropical system is interacting with the climate in a functionally new way for the last 30 yrs? How can we be sure 60 years is sufficient to draw this conclusion (I'm sure you could argue because the climate is relatively stable vs other systems, for example...). The authors seem to posit this is due to CP vs EP, is this the main explanation, what else? And how much of the variability in terrestrial carbon fluxes do these ENSO teleconnections explain? Finally and this may be difficult to address so ignore as you wish ... this study focussed on the tropics, I guess the context I lack is whether this behaviour shown is seen globally (but "we've" only looked at the tropics), or whether this is only seen in the tropics, in which case this is a hugely important story and unique to the tropis...

Response: We appreciate your positive feedback on our study and constructive comments for improvement.

According to your suggestion, we now explore more aspects of models, including simulated photosynthesis and respiration, simulated soil moisture, etc, and show details in your specific comments below.

We now clarify the differences between the interannual variation (IAV) and the trend: "IAV (detrended)" sensitivity of CO₂ growth rate (CGR) and climate allow us to explore carbon-climate relationships because the IAV of CGR clearly stems from climate variability but the rising trend of CO₂ is mainly caused by anthropogenic emissions and other slow varying factors like CO₂ fertilization and nutrient deposition/changes in availability affecting land sinks; "Trend" is to look at the changes in carbon-climate relationships in the context of climate change, which is derived using IAV of CGR and climate during different time periods.

Yes, the big take home message is that tropical water availability increased the control of the interannual variability of tropical land carbon cycle during recent 30-yr under climate change,

which suggests the dominant climate driver of the interannual variation of terrestrial carbon cycle has already shown a tendency to shift from temperature to water. Since the substantial year-to-year variability of land sink arises from the quick response to climate variability, and a 30-yr is usually used as a baseline in climate science, 60 years are sufficient to investigate changes in carbon-climate relationships, but we cannot rule out decadal variability.

Yes, the fact that process-based models do not reproduce this intensified water-carbon IAV coupling makes the thorough elucidation of underpinning processes difficult. We state that shifts in ENSO teleconnections on tropical water storage could be only one mechanism and therefore discussed some other possible mechanisms (Lines 255-260 in the original manuscript). Terrestrial carbon fluxes are driven by terrestrial water availability directly, which in turn are largely controlled by ENSO. Our focus here is tropical terrestrial water availability has an increasing explanation for CGR IAV.

We agree that it is interesting to extend whether intensified water-carbon coupling still works in other regions in future studies.

Reviewer C1.1:

Other thoughts...

Perhaps there isn't the space, but I found myself wanting to first be told how much tropical temperatures vary. We begin by being told there is a correlation between CGR and tropical temperatures. But one naturally assumes tropical temperatures are (relative) less variable than other locations. Would it be helpful to add a sentence with this as the starting point before you discuss correlation?

Response: We appreciate your feedback in highlighting this point and we apologize for the lack of clarity. Because the large IAV of CGR are dominated by exchanges in tropical land-atmospheric carbon fluxes, the tropical temperature is correlated to CGR, despite the lower variability of tropical temperature than other regions (Zeng et al., 2021). We made textual revisions to highlight it.

Reference:

Zeng, X., Reeves Eyre, J. E. J., Dixon, R. D., & Arevalo, J. (2021). *Quantifying the occurrence of record hot years through normalized warming trends. Geophysical Research Letters*, 48, e2020GL091626.

Reviewer C1.2:

While Figure 1a is very interesting, I find myself wondering about what the temperature (water) anomaly means in the context of ecosystem behaviour. To explain, the temperature anomalies are relatively small and represent an annual average. Mechanistically, how much difference would we expect to predict in carbon uptake/respiration for these magnitudes of anomaly? I dare say that we wouldn't actually hypothesise we'd be easily able to separate variations between years, owing to the variability in the timing of how these anomalies would present themselves seasonally as a forcing. I guess, seeing as you have the models, would these patterns look like? I would imagine far, far messier, which one could attribute to poor model skill, but perhaps in reality, also that this reln should be more variable...? While ~18 (roughly by eye) of the biggest CGR anomalies are in the hot & dry, ~9 are not, including two of the biggest anomalies. So, I guess, yes there is a pattern and also, it isn't as simple as the graph is trying to suggest? I don't think the text really reflects what you see by looking at the plot, instead it opts for the simpler summary without pointing to the examples that don't fit the pattern. I think this matters and perhaps strengthens the argument: 1a is showing to me there is still marked IAV and it isn't easily attributable to just been hot & dry or the opposite. By contrast, the rest of figure 1 is nicely showing that despite this variability, the trend is clear. My general take from this figure is that the variability in CGR isn't necessarily well

understood on IAV timescales but we can still see that the role of water availability is driving a background change. I feel like this isn't necessarily the interpretation the authors have though...or at least, I'm not getting this from the text (it may just be me!).

I really like the analysis in 1b-e (which was nicely supported by an alternative approach in the supplement).

Response: We appreciate your acknowledgements of our main finding (Figure 1).

Yes, due to the lack of direct observations of photosynthesis and respiration covering such a long time period (1960-2018), we cannot check their response to climate variations from observations. Following your suggestion, we check the response of photosynthesis and respiration to climate variations in models. In general, compared to ecosystem respiration, the simulated response of ecosystem gross primary production to soil moisture in models is more consistent (Fig.S8 and S9 in the revised manuscript).

Thanks for pointing out some examples in Figure 1 that do not fit the general pattern. We revised the text to clarify this: "*There is also a small proportion of CGR that does not fit the general pattern here (Fig. 1a), suggesting the role of other factors, like the exceptional (non-linear) anthropogenic emissions or ocean carbon sink.*"

Reviewer C1.3:

Lines 43, 47, 50: It would be good to tell the reader how strong the reln between CGR and ENSO is - what is the correlation? Similarly, the correlation for lagged precipitation and CGR? And GRACE and CGR. In each instance the reader is told it is "strong"... These statements appear liberally throughout the text. For example, Zhang et al (<https://onlinelibrary.wiley.com/doi/full/10.1111/gcb.14275>) say ENSO is explaining <30% of productivity (I think from memory) - so with this in mind - is it worth some discussion about other drivers beyond ENSO? It may be that this study isn't breaking this down into EP vs CP, I honestly can't recall - but my general point is with a little editing, the authors could better inform the reader...and I think it would enhance the manuscript.

Response: Following your suggestion, we now add these estimates to lines 43, 47 and 50. As in the previous reply, we discussed other drivers beyond ENSO.

Reviewer C1.4:

With the coupled model analysis - the text isn't clear, during the historical periods, is each model following the same climate forcing patterns, in CMIP6 (presumably not, i.e. for each historical period, each model has its own random sequence of meteorology)? I doubt this will be obvious - can this be explained to the reader in the methods? As such, I cannot possibly see how this comparison makes sense? Forcing the LSMs with the same climate as you see during the period makes more sense, but probing the CMIP models?

Response: We now clarify this point in the Methods. For coupled earth system models, the "historical" scenario is used. Climate and land carbon sinks are simulated with all-forcings including both the natural causes (e.g., volcanic eruptions and solar variability) and human factors (e.g., GHG concentration, aerosols, and land use) over the period 1850–2014. In coupled earth system models, the carbon cycle is coupled to the climate system. We use simulated global NEE from models to represent CGR. Since coupled earth system models are widely used to predict future climate, whether they can reproduce historical climate, land sink and terrestrial carbon-climate coupling is of great interest.

Reviewer C1.5:

I understand that with the multi-models, it is hard to recreate the analysis shown for the obs, but I do feel it would at least be helpful to show each model as a version of Fig 1? I would really like to see a version for NEE for the histograms (even if in the supplement).

Fundamentally, I find the model comparison extremely weak. It just says things are different and we need improvements. This actually entirely unhelpful. How are these results going to motivate the models to improve? Can we at all diagnose something more useful? As I suggested above - what is our expectation re: the variability to these anomalies? For example, the authors hypothesise that a driving factor in the obs is the change in the teleconnections, shift from EP to CP - do we think the obs forcing captures this in the offline met driving? Are the models not capturing the change in coupling because it isn't in the driving data or because the processes are wrong? Given you have also shown the coupled models, then I think you can also answer about whether the coupled models are capable of simulating the change in CPvsEP - presumably not... Is there one model or a few models that do simulate the CP vs EP change and do they better simulate the change?

Response: We appreciate your constructive suggestions on extending the model part in this MS. Similar to Fig.1 for observations, we actually produced the Figures for models. Fig. S4 in the supplementary info showed detailed temporal dynamics of interannual (partial) correlations between tropical soil moisture and global NEE from most models.

Following your suggestion, we explore more aspects of models. In our case, the ability of models to reproduce observed terrestrial water-carbon coupling depends not only on the simulation of terrestrial water availability but also on process representations of the carbon cycle 's response to climate. That's also why the simulated land sinks are largely divergent among offline models using the same climate forcing. We further find models might not represent the carbon cycle processes well because the modelled carbon-water coupling is stable over time, regardless of the differences among simulated soil moisture. Specifically, simulated soil moisture from all offline models can generally capture teleconnections to CP and EP ENSO (Fig. S7 in revised MS), while it is documented that the current generation of coupled models do not represent the ENSO diversity well, partly due to dry and cold biases in the equatorial central Pacific (*Ham et al., 2012; Timmermann et al., 2018*); The dominant spatial patterns of simulated soil moisture anomalies are largely divergent among models (Fig. S5 and S6 in revised MS). We now add new results and discussions to the revised manuscript.

Reference:

Ham, Yoo-Geun, and Jong-Seong Kug. "How well do current climate models simulate two types of El Niño?." Climate dynamics 39.1 (2012): 383-398.

Timmermann, Axel, et al. "El Niño–southern oscillation complexity." Nature 559.7715 (2018): 535-545.

Reviewer C1.6:

As for the VOD - I'm have no expertise in these data, but I wonder if it makes sense the differences should be so pronounced!?? If we think VOD is a proxy for aboveground biomass, wouldn't we predict a smaller change in the stock vs the change in fluxes? Yet the different in histograms in S7 is far greater between periods! I presume this have accounted for land use change?

Response: We appreciate your valuable suggestions. Following *Fan et al., 2019*, we use VOD derived AGC (AGC_{VOD}) fluxes and raw VOD fluxes (the first-order differences of AGC_{VOD} and raw VOD), which are not a stock but only a part of net land carbon fluxes (IAV of biomass sink change). However, according to Reviewer#2 comment, we now reduce this part because the underpinning driver of AGC_{VOD} IAV could also be soil moisture, in addition to biomass. VOD is also sensitive to plant water content, which is itself sensitive to soil moisture IAV. Therefore, interpreting IAV of AGC_{VOD} as biomass IAV alone should be taken with care and this topic is

still in debate. Nonetheless, this is not the concern of our study and still provides independent evidence to support our main findings.

Reference:

Fan, L., Wigneron, J. P., Ciais, P., Chave, J., Brandt, M., Fensholt, R., ... & Peñuelas, J. (2019). Satellite-observed pantropical carbon dynamics. Nature plants, 5(9), 944-951.

Reviewer C1.7:

Finally, it shouldn't need to be said but: "Codes are available under reasonable requests" isn't acceptable. I urge the authors to put their code into a repository and make it freely available, regardless of what the journal's policy is.

Response: We appreciate your valuable suggestion. We now add it.

Reviewer #2 (Remarks to the Author):

Reviewer C2.0:

This paper investigates the temporal variability of the correlation between the carbon growth rate and both water availability and temperature variations and finds the former is strengthening. The paper further argues that ENSO changes and changes in spatial patterns of water storage (TWS) anomalies underly this pattern. Understanding how climate affects the carbon growth rate (CGR) through its effect on the terrestrial carbon cycle is a fundamental component of understanding climate change, so the paper's advances will be of interest to Nature's readership. The explanation the authors provide based on changes in the degree of spatial compensation is compelling. I also commend the authors for illustrating a key limitation in the Wang et al Nature 2014 analysis. However, I have several fundamental concerns about the robustness of the results, as well as the novelty of some of them.

Response: We appreciate your overall acknowledgements of our study and insightful comments for improvement.

Reviewer C2.1:

1) The authors investigate the strength of the relationship between TWS and CGR and refer to this in many locations as the 'sensitivity' (e.g. lines 126, 128, etc). However, the sensitivity of the CGR-TWS relationship, and the quantity of interest for understanding the CGR, is the slope, not the correlation coefficient of this relationship. This quantity does not seem to be investigated anywhere.

Response: We appreciate your feedback in highlighting this point and we apologize for the lack of clarity. We actually already did this sensitivity analysis (Fig. S2 and Lines 126-138 in the original manuscript). This sensitivity analysis was performed in the same way as Wang et al., Nature 2014 mentioned by the referee. In addition, we replaced concurrent precipitation with terrestrial water storage in the sensitivity analysis.

We now revised the text to clarify this point: "The sensitivity of CGR IAV to climate is further determined using multiple linear regressions with tropical temperature and tropical water storage, defined as the slope of the regression between CGR and climate, with both variables detrended".

Reviewer C2.2:

2) The analysis relating water storage to ENSO type in Fig. 2d,e considers only the overall spatially averaged water anomaly, if I understand the analyses correctly. However, while this ENSO analysis is convincing, and the analysis in Figure 2a,b,c is convincing regarding the spatial coherence shift, it does not explicitly show that the change in spatial coherence of the water storage anomaly with time is directly attributable to the ENSO shift, as directly linked in lines 163-165 and implied elsewhere. Could these two factors not be independent?

Response: We appreciate your feedback in highlighting this point and we apologize for the lack of clarity. We actually already showed spatially explicit details of ENSO teleconnections on water storage anomaly (**Fig. S3 and Lines 168-169 in the original manuscript**).

We now revise the text and integrate this Figure to Figure 2 to clarify this part: "Moreover, it is apparent that much larger areas of tropical water storage anomalies are correlated to CP ENSO compared with EP ENSO in the recent 30-yr (Fig. 2c, d)".

Reviewer C2.3:

3) The Humphrey et al, Nature, 2018 paper cited here already showed that DGVMs are unable to capture the CGR vs. water storage and CGR vs. temperature correlations. Since the process representations of DGVMs have much in common with uncoupled models, the failure of these models to capture the right trend in correlation when they can't get the overall average correlation right (as shown in Figure 3) is not really a big surprise/particularly novel. The discussion of missing processes also contains significant duplicity with that in Humphrey et al, 2018. This should be at least acknowledged.

Response: We appreciate your feedback in highlighting this point and we apologize for the lack of clarity. *Humphrey et al., 2018* showed DGVMs cannot capture CGR vs. global water storage correlations (*Fig. 2b and Fig. 4a in Humphrey et al., 2018*) but DGVMs are able to capture CGR vs. tropical water storage correlations well (**Fig. 2d in Humphrey et al., 2018**), in which the investigated time period spans from 1980 to 2016.

Here we focus on CGR vs. tropical water storage relationships. Our contribution is the first one (to our knowledge) to demonstrate the increasing tropical water control on interannual variations of CGR from 1960 to 2018 based on observations, while both offline models and coupled models cannot capture this. This was not reported by *Humphrey et al., 2018*. We acknowledge *Humphrey et al., 2018* in the mentioned discussion part.

Reviewer C2.4:

4) As the authors mention on line 241, VOD is sensitive to canopy water content. Thus, its interannual fluctuations (and thus correlations to interannual CGR changes), can be potentially driven by changes in biomass or by changes in relative water content. The analysis the authors show in Figure S10 only investigates the *spatial* correspondence between VOD and AGC, not the correspondence of temporal variations. It is therefore not sufficient to demonstrate the robustness of the lines 234-260. In fact, as shown in detail in Konings et al., GRL, 2021, VOD interannual fluctuations are likely a poor proxy for AGC interannual variability because of the strong influence of interannual variations in relative water content. Because relative water content changes are likely to be significantly affected by changes in water storage (and thus the water storage-CGR coupling investigated here), this analysis is unlikely to be robust and should be removed. Instead, why not simply look at GRACE and the CGR directly for either GRACE in semi-arid regions vs. GRACE in moist regions? This would be a more explicit, data-driven demonstration of the role of semi-arid regions. Although the coarse resolution of GRACE may add some noise, this analysis would likely be significantly more robust than the one currently in lines 234-260.

Response: We appreciate you for providing this interesting study from *Konings et al., 2021*. *Konings et al.* assumed that the *Xu et al.* AGB annual product was “truth” and inferred water content sensitivities. We argue that this is a strong hypothesis since true AGB IAV is not known. Zhang et al. modeled VOD from water potential proxies and Normalized difference vegetation index (NDVI) and found that leaf water potential could only explain 17% of the variance of VOD (AMSR-E) thus a dominant fraction being explained by AGB. We found this topic is still in debate and are aware that interpreting IAV of VOD-derived AGC as biomass IAV alone should be careful. The VOD part aims to provide independent evidence in this study. We now cite *Konings et al., 2021* and clarify this point in the revised MS. We now reduce the VOD part and move it to supporting info.

Reference:

Xu, L., Saatchi, S. S., Yang, Y., Yu, Y., Pongratz, J., Bloom, A. A., ... & Schimel, D. (2021). Changes in global terrestrial live biomass over the 21st century. Science Advances, 7(27), eabe9829.

Zhang, Y., Zhou, S., Gentine, P., & Xiao, X. (2019). Can vegetation optical depth reflect changes in leaf water potential during soil moisture dry-down events?. Remote Sensing of Environment, 234, 111451.

More minor comments:

* Lines 43-45: Use of the past passive voice here is confusing, as it is unclear whether you are talking about your own results or those of others.

Response: Corrected.

* Line 86: The more recent periods (e.g. the 1994-2018 period mentioned in line 86 and others shortly before it), overlap with the decadal ‘warming hiatus’ period, which is at least partially caused by internal variability. See, e.g. Medhaug et al, 2017. How does this affect our expectations for whether this behavioral shift is likely to continue well into the future? At the very least some discussion on this general topic seems warranted.

Response: We cite *Medhaug et al, 2017* and discuss this general topic as follows:

“The most recent 30-yr (1989-2018) overlaps with the decadal ‘global warming hiatus’ (1998-2012) in which natural internal variability, like ENSO, might play a part (Medhaug et al, 2017). The effects of internal variability should not change the increased tropical water control on CGR IAV in the future because we have already taken ENSO into account but deserve more investigations.”

* Figure 1b: It's a bit difficult to see that the blue axis are negative correlations. Perhaps this can be reflected in the axis label instead?

Response: Corrected by enlarging the axis Fontsize.

* What spatial and temporal space is Figure 2e calculated at?

Response: Water storage covers the tropical vegetated land areas (24°N and 24°S), as in *Humphrey et al. 2018*. The time period spans from 1989 to 2018.

References:

Humphrey, V., Zscheischler, J., Ciais, P., Gudmundsson, L., Sitch, S., & Seneviratne, S. I. (2018). Sensitivity of atmospheric CO2 growth rate to observed changes in terrestrial water storage. Nature, 560(7720), 628–631. <https://doi.org/10.1038/s41586-018-0424-4>

Konings, A. G., Holtzman, N. M., Rao, K., Xu, L., & Saatchi, S. S. (2021). Interannual Variations of Vegetation Optical Depth are Due to Both Water Stress and Biomass Changes. *Geophysical Research Letters*, 48(16), e2021GL095267. <https://doi.org/10.1029/2021GL095267>

Medhaug, I., Stolpe, M. B., Fischer, E. M., & Knutti, R. (2017). Reconciling controversies about the 'global warming hiatus.' *Nature*, 545(7652), 41–47. <https://doi.org/10.1038/nature22315>

Reviewer #3 (Remarks to the Author):

Reviewer C3.0:

Liu et al. investigated the temporal trend of the relationships between tropical water and the interannual variability (IAV) of atmospheric CO₂ growth rate during 1960-2018, primarily using reconstructed total water storage data, precipitation from CRU, and the CO₂ growth rate from NOAA surface sites. They concluded that the tropical water availability is increasing controlling the IAV of the atmospheric CO₂ growth rate and thus the tropical terrestrial biosphere carbon cycle. The paper further analyzed CMIP models and several offline terrestrial biogeochemical models, and found that these models could not reproduce the intensified water - carbon coupling. As uncertainty in carbon-climate feedback is one of the leading causes in the uncertainties of climate projections, the topic of this study is important. However, I have a few major concerns. Please see below for my detailed comments.

Response: We appreciate your acknowledgements of the importance of our study and your insightful comments.

Reviewer C3.1:

1) The uncertainty for the linear fitting lines in Figure 1b and 1d do not seem to include the uncertainty in the original dataset. As total water storage (TWS) data is from modeling estimates, it is important to consider its uncertainties in the linear fitting. Figure S2 shows the sensitivities of CGR to climate variations, which do show a very large uncertainty.

Response: We appreciate your feedback in highlighting this point. The uncertainties of the interannual correlations of CGR-Water arising from six different configurations of terrestrial water storage datasets is now added to the Figure 1b and 1d, which are quite small. The uncertainty mentioned by the referee (shading areas) in Figure S2 refers to the 95% confidence level of estimates using the bootstrapping approach (randomly selecting a subset of years in each time window); Figure 1c and 1d already showed the distribution of interannual carbon-climate correlations using the same bootstrapping approach.

Reviewer C3.2:

2) The conclusion of increasing tropical water control on interannual CO₂ growth rate is not supported by the analysis. Controlling for temperature effect, the absolute value of $R(\text{WS}, \text{CGR} | T)$ has the same magnitude in late 1960s (0.4) as in early 2000s (-0.4) (Figure 1d), only that they have different sign. Thus, the tropical water storage has the similar magnitude of control on CGR in the late 1960s as in early 2000s when controlling temperature, but just in the opposite direction. TWS affects both plants growth and ecosystem respiration, and the net of which controls CGR. The effect of TWS on plants growth and respiration could have changed due to change of climate, which needs further investigation. In Figure 2, the authors argued that the spatial coherence of WS anomaly is the reason for the changes in $R(\text{WS}, \text{CGR} | T)$. But since $R(\text{WS}, \text{CGR} | T)$ in late 1960s has the

similar magnitude as that in 2000s, following the same argument, it also requires some coherence in WS anomaly.

Response: We appreciate your feedback in highlighting this point. As the reviewer's 3rd comment below, the robust finding needs cross-validations and we do not base our findings from two data points. Therefore, we use several different datasets and the bootstrapping approach. To demonstrate the robustness of our findings, we made a new table **Fig. R3-1**, which summarized the estimates of $R(WS,CGR|T)$ from the first 30-yr (1960-1989) to the recent 30-yr (1989-2018) using 5000 bootstrapping repeats from several datasets. Note that we include the Global Precipitation Climatology Centre (GPCC) precipitation dataset here and explain the reason in the response to 3rd comment below. As we stated in the original manuscript, we use CGR to represent the terrestrial land-atmosphere carbon fluxes. To verify that the IAV in CGR does not primarily originate from fossil fuel burning and cement production, land-use change, and ocean uptake, we also use the residual land sink (RLS) estimated from the Global Carbon Budget instead of CGR. It is apparent that the large positive values of $R(WS,CGR|T)$ do not hold when using other climate datasets or RLS. **Table R3-1** robustly shows that tropical water-carbon coupling significantly increased in the recent 30-yr compared to the first 30-yr. We now add this table to the revised Manuscript (MS).

Table R3-1. Summary of temporal dynamics of interannual partial correlations of carbon fluxes to land water after controlling temperature (mean \pm s.d.) during 1960-2018. Estimates are derived from 5000 bootstrapping repeats from different datasets.

Data Sources	Time period		
	First 30-yr (1960-1989)	Recent 30-yr (1989-2018)	Significant level (two-sample t-test)
Carbon (CGR) Ta (CRU); Water (GRACE-REC)	0.13 \pm 0.22	-0.28 \pm 0.21	Significant, P<0.01
Carbon (RLS) Ta (CRU); Water (GRACE-REC)	-0.02 \pm 0.19	-0.34 \pm 0.20	Significant, P<0.01
Carbon (CGR) Ta (CRU); Water (CRU)	0.00 \pm 0.25	-0.42 \pm 0.18	Significant, P<0.01
Carbon (CGR) Ta (Berkeley); Water (GPCC)	-0.07 \pm 0.23	-0.35 \pm 0.21	Significant, P<0.01

Reviewer C3.3:

3) The data quality of both reconstructed water storage (TWS) data and precipitation data from CRU are questionable over the tropics. Figure 5 in Humphrey and Gudmundsson (2019) show that the correlation between reconstructed TWOS and the original dataset is close to zero over most of the tropical Africa, and less than 0.3 over most of the tropical South America. Figure 6 in the same paper shows that the Nash-Sctcliffe efficiency is less than 0.2 (closer to 1 corresponds to better performance) over most of the tropical region. It is unclear to me why Figure S8 shows much higher correlation than in Humphrey and Gudmndsson.

Even though Figure S8 shows much higher correlation than in Humphrey and Gudmundsson, Rec-TWS does show poor performance of over tropical Africa, which is critical in the spatial pattern analysis in Figure 2a and b.

Response: We appreciate your highlighting of this point and we apologize for the lack of clarity. Unlike Figs. 5 and 6 in Humphrey and Gudmundsson et al., 2019 which show prediction skills at the **monthly timescale (detrended and deseasonalized)**, our evaluation confirms the quality of reconstructed water storage is very good at the **yearly timescale (detrended)** (Fig. S8b). We used the latest version 3 of GRACE-REC and the ensemble mean of reconstructions. Since we focus on interannual correlations, Nash–Sutcliffe efficiency is not our concern here.

We acknowledge that the quality of reconstructed water storage shows poor performance in central Africa, even at the yearly timescale. First, we clarify the aim of Fig. 2a and 2b (leading EOF of water storage anomalies) is to illustrate parts of the spatial coherence of water storage anomalies because the leading EOF only explain about 18.9% and 18.5% of spatial variance in 1960-1989 and 1989-2018, respectively (See **Line 150** and **Lines 381-383 in the original MS**). The total degree of spatial water storage anomalies coherence was quantified by the metric based on the larger covariance matrix (**Fig. 2c** and **Lines 386-398 in the original MS**). Therefore, the spatial coherence metric is most critical in the spatial coherence analysis here. To test the robustness of our spatial coherence analysis, we perform a new set of analyses by excluding central Africa (Outlined by the black rectangle in **Fig. R3-1a**). **Fig. R3-1b** confirms that excluding these areas would not bias the temporal dynamics of the spatial coherence metric and the result here, despite all values being a bit higher. We also removed Fig. 2a and 2b from Fig. 2 to avoid uncertainties.

Fig. R3-1 (a) Spatial distribution of the interannual correlation of original GRACE-TWS to GRACE-REC at the yearly scale during 2002-2016. A black rectangle indicates the areas of central Africa to be excluded from the analyses, covering 10°W to 38° E, 9°S to 10° N; **(b)** Comparison of spatial coherence metric from total tropical lands and from tropical lands without central Africa. Each dot indicates a 25-yr moving window.

Second, to test the robustness of our main finding (increasing tropical water control on CGR IAV), we use GRACE-REC without central Africa to perform the analyses again. **Fig. R3-2** below confirms that our finding is robust. This can be expected because tropical aggregated GRACE-REC reproduces the original TWS very well.

In summary, our main finding of increasing tropical water-carbon coupling is robust. We now add the robustness test by excluding central Africa to the revised manuscript.

Fig. R3-2 Same as Fig. 1c and e, but using water storage excluding central Africa that covers 10°W to 38° E, 9°S to 10° N.

Reviewer C3.4:

CRU is also a reconstructed dataset based on station data. Figure 11 in Harris et al. (2020) shows that correlation coefficient from cross-validation for precipitation is less than 0.3 over most of the tropical continents. In addition, the number of observation stations is very sparse over the tropics (Figure 1 in Harris et al., 2020). Coupled with the changing number of stations and the heterogeneity of the precipitation in general, the variability analysis with the dataset is questionable. In Figure S1, there is a big jump in the $R(\text{pr}, \text{CGR})$ changing from close to 0 to 0.3 in 1990s. If it were not due to the change of data quality of change of observation stations, what could have caused that? In Figure S1c and Figure 1d, there is also a jump in 1995.

Response: We appreciate your feedback in highlighting this point and we apologize for the lack of clarity. We are confident that the following evidence can address the mentioned concern.

First, to test the reliability of the **interannual variability (IAV) (detrended)** of CRU tropical precipitation, we utilize **satellite observations** of tropical precipitation from Tropical Rainfall Measuring Mission (TRMM). TRMM was a research satellite launched in 1997 and measured precipitation from space using several space-borne instruments, such as precipitation radar and TRMM microwave imager. **TRMM provides valuable precipitation measurements of the entire tropics, especially over regions without good gauge coverage.** As shown in **Fig.R3-3**, the consistency of tropical precipitation IAV between CRU TS4.03 and TRMM 3B43

confirms the reliability of CRU tropical precipitation. In addition, we actually already tested the reliability of CRU tropical precipitation by comparing it with independent satellite observations of tropical water storage during 2002-2018 (Fig. S9a or Table S1 in the original MS). For complement, we reproduced Fig.R3-4 here to demonstrate the signals of CRU precipitation IAV are reliable. Unlike Fig. 11 in Harris et al., 2020 mentioned by the reviewer which is based on individual interpolated values of **monthly** precipitation (time series length could be only 20 months according to Harris et al., 2020), we show the signals of precipitation variability from gauge stations are very reliable at the **yearly time scale** (our interests here).

Fig. R3-3 Interannual variation in detrended anomalies of tropical land 6-month lagged-precipitation from CRU and TRMM during 1999-2018.

Fig. R3-4 Interannual variation in detrended anomalies of tropical land 6-month lagged-precipitation from CRU and tropical terrestrial water storage during 2002-2018.

Second, to test whether our results are sensitive to the changes of gauge stations, we use the reference Global Precipitation Climatology Centre (**GPCC**) **precipitation dataset** used by Harris et al., 2020 to perform the analyses again. Note that Harris et al., 2020 used GPCC as the reference object because it has the most gauge stations. **Fig. R3-5 indicates that the number of GPCC's tropical gauge stations is even larger than that of CRU's global stations.** In fact, compared to the 2000s to 2010s (in which satellite observations confirm the reliability of CRU tropical precipitation IAV), the spatial density of gauge stations was much larger in the 1960s-1990s (**Fig. R3-5 and Fig.1 in Harris et al., 2020**), and tropical precipitation IAV data could be even more reliable. The drop of station coverage in recent years is largely due to the delay in assembling records. As expected, **Fig. R3-6** confirms the consistency of tropical precipitation IAV between CRU and GPCC. **Fig. R3-7** further confirms that our main findings are robust when replacing CRU precipitation with GPCC precipitation,

suggesting our results are robust to the changes of gauge stations. In addition, the jumps of water-carbon coupling mentioned by the referee are interesting and might indicate the abrupt increase in tropical water-carbon coupling, but they would not impact our main finding and these jumps are not consistent across datasets. For instance, the jump of $R(\text{CGR}, \text{LagP}|\text{CGR})$ from 0 (1979-2003, 1991 as the central year) to 0.3 (1980-2004, 1992 as the central year) does not exist in Fig. 1d (using WS rather than LagP) and Fig. S1c (using RLS rather than CGR). We further check this jump of $R(\text{CGR}, \text{LagP}|\text{CGR})$ from 1979-2003 to 1980-2004 could be due to the difference of the year 1979, and $R(\text{CGR}, \text{LagP}|\text{CGR})$ without the year 1979 still increased from -0.23 (first 30-yr) to -0.58 (recent 30-yr).

In summary, we are confident that our main finding of increasing tropical water-carbon coupling is robust. We now add the validation of CRU precipitation datasets and the robustness tests to the revised manuscript.

Fig. R3-5 Temporal coverage of the number of gauge stations in GPCC and CRU precipitation dataset

Fig. R3-6 Interannual variations in detrended anomalies of tropical land 6-month lagged-precipitation from (a) CRU and (b) GPCC.

Fig. R3-7 Same as Figure 1d, e, but replacing water storage with 6-month lagged precipitation from GPCC.

Humphrey, V. and Gudmundsson, L.: GRACE-REC: a reconstruction of climate-driven water storage changes over the last century, *Earth Syst. Sci. Data*, 11, 1153–1170, <https://doi.org/10.5194/essd-11-1153-2019>, 2019.

Harris, I., Osborn, T.J., Jones, P. et al. Version 4 of the CRU TS monthly high-resolution gridded multivariate climate dataset. *Sci Data* 7, 109 (2020). <https://doi.org/10.1038/s41597-020-0453-3>

Reviewer C3.5:

4) The proposed mechanism for the increasing carbon-water coupling and the analysis of terrestrial biosphere models (TBMs) is not coherent. The study proposed that the changes in ENSO effect on tropical TWS anomaly strengthen tropical water control on CGR. If that the case, the offline TBMs should be able to reproduce the increasing control of water on CGR, since these models used reanalysis as forcing.

Response: We appreciate your feedback in highlighting this point and we apologize for the lack of clarity. The same reanalysis climate forcing cannot ensure the same carbon cycle response in different TBMs; this is why the simulated land sinks are largely divergent among TBMs using the same reanalysis forcing. The ability of TBMs to reproduce observed terrestrial water-carbon coupling depends not only on the simulation of the terrestrial water availability but also on process representations of the carbon cycle's response to climate. First, although precipitation could be similar, simulated soil moisture is the direct water pool for plant water uptake in TBMs. Soil moisture in current TBMs is also simulated and is thus not equivalent to terrestrial water storage. Second, our results, in turn, indicate TBMs might not represent the latter part well (how the carbon cycle response to climate in models) because the modelled carbon-water coupling is stable, regardless of the differences among simulated soil moisture. This is consistent with the previous finding by *Wang et al., 2014 Nature*, which found that four out of five TBMs models (previous version) driven by the same reanalysis do not reproduce the temporal changes in CGR-temperature coupling over time during 1960-2011. We now add these clarifications to the revised MS. We also discussed the potential reasons for model performances (Lines 212-217 in the original MS).

Reviewer Reports on the First Revision:

Referees #1 and #3 were unable to review the revised version of the paper. Referee #4 and #5 were asked to assess the authors' responses to the points raised by referee #1 and #3 instead.

Referees' comments:

Referee #2 (Remarks to the Author):

I thank the authors for addressing my comments from the earlier review. However, I am not fully convinced by their responses along two of my previous major concerns:

1) Sensitivity estimates through correlation vs. slope: Figure S2 seems to show a much less dramatic increase in water anomaly control on the CGR than the correlative analysis does. For example, the slope with respect to the water variable (panel a) seems to possibly stall or even increase after around 1997. This is particularly relevant because the y-axis changes sign. It's hard to read because of the overly broad y-axis limits, but it looks like the magnitude at the end of the period is actually smaller than at the beginning. Thus, while squinting at panel a shows a general decrease, the temporal pattern is much less clear than the correlative analysis. This is somewhat troubling, if it is ultimately this sensitivity that is the quantity of interest for the reader who is trying to better understand what controls the CGR under future climates. The fact that the red line for temperature (panel b) has even less clear of a trend only further adds to this concern.

2) Spatial coherence of water storage anomaly and ENSO. I appreciate that the authors have moved the correlation maps to the main figure 2c,d. However, I disagree that these maps show that the CP ENSO patterns explain the changing spatial coherence of the WS anomaly. The spatial coherence is not calculated in these maps. I agree with the authors' interpretation of Figure 2c,d that the correlations with the water storage anomaly at each location are higher for the CP SSTs (panel d) than for the EP SSTs (panel c). But this is to be expected when we know that the tropically averaged water storage anomaly correlates well with the CP SSTs, which was already shown in panels a and b of figure 2. So with panels 2a-d you conclusively show that the water storage anomaly varies with CP. But this says nothing about how the spatial coherence is changing because of ENSO (the intriguing topic of panel 2e), and therefore it does not provide any support for the authors' claim that ENSO patterns might be affecting the change in the way that interannual water storage anomalies affect the CGR. This in turn leaves me wondering what Figures 2a-d add to the manuscript.

Minor:

Figure S4: typo in label

Referee #4 (Remarks to the Author):

I've been asked to review the author's replies to referee #1. I have read through the rebuttal letter as well as the manuscript and supplementary figures. Going through each comment and response, I can confirm that the authors have done a good job in responding to the comment and making the necessary changes in the paper.

Overall, the reviewer did not have any “fatal” concerns with the original manuscript. Rather, the reviewer wanted to see: 1) a better use of the model data, 2) a better clarification of what was meant by IAV and trend, and more 3) more clear take-home messages.

Many of the point by point comments concern #1 and the authors have done well at addressing each of these issues regarding the use in models. While I agree with the reviewer that this kind of result (i.e., the models don’t get it) is usually not very helpful, the authors have added additional figures and discussion concerning why the models might be wrong.

On #2 and #3, the main take home message of the paper is clear, i.e., that the coupling between tropical water availability and CGR appears (note that correlation and regressions used in the paper don’t show causation so the “coupling” may be due to entirely different reasons) to have increased (trend) over the last 60 years for which we have data. The result that the carbon cycling in the tropics is also controlled by water availability has already been noted in several recent studies, but the result that this control has changed over the last 60 years possibly related to the spatial shifts in ENSO events seems to be the paper’s novelty. It is up to the editor to decide whether that warrants a space in their prestigious publication.

Referee #5 (Remarks to the Author):

Dear editor,

Because you said you are not asking for a full review, but to assess the authors' replies to the referee #3. The question of “Reviewer C3.2” is important. The datasets used in the author’s main text show that the tropical water storage has comparable but opposite control on CGR in the 1960s and 2000s with the partial correlation coefficients about 0.4. Authors suggest that water control from 0.4 in 1960s to -0.4 in 2000s shows the increasing coupling between tropical carbon cycle and soil water, owing to the spatial coherence of WS anomaly. Referee#3 argued that if -0.4 in 2000s is owing to the increase in spatial coherence of WS, 0.4 in 1960s should also require the coherence in WS anomaly. In the reply, authors did not directly address whether coherence of WS can explain the 0.4 in 1960s based on their datasets in the main text. However, they used other datasets (GPCC, CRU precipitation) in the reply file to show that 0.4 cannot hold. So what’s the reason of 0.4 in 2000s?

Another concern is that these partial correlation coefficients were calculated in a 25-yr moving window. In Figure 1d (main text), a lot of correlation coefficients ranging from 0.4 in 1960s to -0.4 in 2000s have no statistical significance (close to zero). Even that they use GPCC/CRU in Table R3-1 to calculate the partial correlation coefficients, all the correlation coefficients in the first 30-yr are not significant.

In the t-test for the correlation and regression, it needs the degree of freedom (dof). The moving window will cause high auto-correlation, making the dof low (so dof is not equal to N-2). They need to re-calculate the dof to calculate the significance (not just the p-values directly given by python or R functions).

Author Rebuttals to First Revision:

We thank the editor and the reviewers for their insightful comments and suggestions. The manuscript has been substantially revised following the reviewers' comments. **Most importantly, we highlight with new evidence that the results are robust. Correspondingly, we replace the original Figure 1 and Figure 2 with new ones and add a new Table to elaborate the full evidence. We also modify the title to "Increasing negative coupling between tropical water and interannual CO₂ growth rate".**

Please see our detailed response below. The original comments are displayed in black, and our responses are in blue.

Reviewer' comments:

Reviewer #2 (Remarks to the Author):

C2.0:

I thank the authors for addressing my comments from the earlier review. However, I am not fully convinced by their responses along two of my previous major concerns:

Response: We appreciate that the reviewer is satisfied with how we addressed most previous comments. We address the remaining two comments below.

C2.1:

Sensitivity estimates through correlation vs. slope: Figure S2 seems to show a much less dramatic increase in water anomaly control on the CGR than the correlative analysis does. For example, the slope with respect to the water variable (panel a) seems to possibly stall or even increase after around 1997. This is particularly relevant because the y-axis changes sign. It's hard to read because of the overly broad y-axis limits, but it looks like the magnitude at the end of the period is actually smaller than at the beginning. Thus, while squinting at panel a shows a general decrease, the temporal pattern is much less clear than the correlative analysis. This is somewhat troubling, if it is ultimately this sensitivity that is the quantity of interest for the reader who is trying to better understand what controls the CGR under future climates. The fact that the red line for temperature (panel b) has even less clear of a trend only further adds to this concern.

Response: We appreciate the reviewer's feedback in highlighting this point and apologize for the lack of clarity in the previous manuscript. We agree that estimates of correlation and sensitivity and their agreement are crucial here. **We elucidate this point by performing a new comprehensive analysis and adding a new Table and a new Figure.**

To demonstrate the robustness of our findings, we now add new analyses to construct the full picture of the evidence and create a new **Table R2-1** and **Figure R2-1**, which summarize the changes in interannual sensitivity and correlation of CGR to terrestrial water from the prior 30-year to the recent past 30-year between 1960 and 2018. New analyses are listed below:

- 1) **We now add estimates on the sensitivity of CGR to LagP (lagged precipitation), in alignment with correlation estimates.** In our previous MS, we followed previous studies (e.g., *Wang et al., 2014 Nature*) in estimating the sensitivity of CGR to concurrent precipitation rather than LagP for comparisons and now remove it to avoid confusion, since LagP was already identified as a better estimate of water availability of relevance to tropical land sink on interannual scale (Method in the MS).

- 2) **We now add the univariate regression in which CGR is the function of terrestrial water alone to avoid possible underestimations of the interannual sensitivity of CGR to terrestrial water variations.** This is supported by *Humphrey et al., 2021 Nature*. This study utilized model factorial experiments to demonstrate that removing soil moisture interannual variability suppresses land carbon uptake variability by about 90%, while tropical mean temperature remains unchanged (*Extended Data Fig. 10 in Humphrey et al., 2021 Nature*). Models do not reproduce the increasing water-carbon coupling over time but roughly capture the sign and strength of long-term tropical water-carbon coupling during 1960-2018, thus providing valuable insights into underpinning processes. Therefore, *Humphrey et al. 2021* suggested that tropical annual mean temperature might not represent a mechanistic climatic driver for land carbon uptake variability but could be a confounding factor.

- 3) **We now add a new method, ridge regression, to perform the bivariate regression analysis to reduce biases from high collinearity between predictors.** Ridge regression is a common technique to estimate the coefficients of multiple-regression models in scenarios when predictor variables are highly collinear (*Hoerl et al., 1970; Hastie et al., 2009*). It reduces issues with overfitting through shrinkage of regression coefficients and has been widely used in the field of climate and carbon science (*Peng et al., 2013; Sippel et al., 2021*). In our case, we now realize that regression coefficients obtained through the ordinary least squares (OLS) fit could be biased because of the collinearity between water and temperature (the correlation between WS and T is about -0.83).

Table R2-1 and Figure R2-1 show that both interannual sensitivity and correlation of CGR to terrestrial water variations estimated from all approaches are significantly different between the two 30-year periods (based on the Wilcoxon signed-rank test, $P < 0.01$) and become more negative over time, despite the specific magnitude and unit of each metric depends on the chosen approach. For metrics considering tropical annual mean temperature, the probability that they are different from 0 at the prior 30-year is low (26.8%) but very high at the recent past 30-year (96.7%). For metrics without considering tropical annual mean temperature, metrics are significantly different from 0 at two sub-periods ($P < 0.01$) and consistently become more negative. For instance, the univariate interannual sensitivity of CGR to tropical WS and LagP increase by about 35% on average from the prior 30-year to the recent past 30-year. **These results robustly confirm that both sensitivity and correlation estimate consistently has become increasingly negative in the recent past (1989-2018) compared to prior climate conditions (1960-1989). We include all these results since they allow for a transparent discussion of uncertainties regarding confounding water-temperature coupling and do not affect our conclusions.** In the revised Manuscript (MS), we now add estimates of sensitivity by replacing Figure R2-1 with original Figure1 and adding Table R2-1. **We now also revise the MS title to “Increasing negative coupling between tropical water and interannual CO₂ growth rate” and other statements throughout the text.**

Table R2-1. Summary of the interannual sensitivity and correlation of CGR to terrestrial water variations during the prior 30-year (1960-1989) and the recent past 30-year (1989-2018). Estimates are derived from 5000 bootstrapping repeats by randomly selecting years without volcano perturbations in each sub-period. The mean and one standard deviation are presented for the metric. The probability that the metric is different from 0 and the corresponding P value are computed by inverting the corresponding confidence intervals. For instance, the probability that the metric is different from 0 is 96% suggests that 96% confidence level of the metric does not include 0 and the corresponding P value is 0.04.

Metric	Method	Proxy of Water	Metrics during first 30-year	Probability that Metrics during first 30-year is different from 0	Metrics during recent 30-year	Probability that Metrics during recent 30-year is different from 0	Test whether the Means of Metrics during the two 30-year periods is equal (Wilcoxon signed-rank test)
Sensitivity	Univariate (OLS)	WS	-0.95 ± 0.27**	99.9%	-1.26 ± 0.23**	99.9%	Not, P<0.01
		LagP	-0.24 ± 0.07**	99.4%	-0.33 ± 0.05**	99.9%	Not, P<0.01
	Bivariate (Ridge)	WS	-0.05 ± 0.33	15.6%	-0.66 ± 0.26**	98.9%	Not, P<0.01
		LagP	-0.06 ± 0.1	43.4%	-0.22 ± 0.05**	99.9%	Not, P<0.01
	Bivariate (OLS)	WS	0.33 ± 0.55	43.4%	-0.68 ± 0.44*	90.4%	Not, P<0.01
		LagP	-0.03 ± 0.13	4.9%	-0.24 ± 0.07**	99.7%	Not, P<0.01
Correlation	R(W,CGR)	WS	-0.53 ± 0.14**	99.9%	-0.71 ± 0.1**	99.9%	Not, P<0.01
		LagP	-0.58 ± 0.15**	99.2%	-0.80 ± 0.06**	99.9%	Not, P<0.01
	R(W,CGR T)	WS	0.13 ± 0.21	44.5%	-0.36 ± 0.20*	90.4%	Not, P<0.01
		LagP	-0.04 ± 0.24	2.7%	-0.56 ± 0.14**	99.6%	Not, P<0.01
				55.3% (On average)		97.9% (On average)	

Note: ** and * indicate a significant sensitivity or correlation at P<0.05 and P<0.1, respectively.

Figure R2-1. Tropical land climate-carbon interannual relationships during 1960-2018. **a.** Yearly tropical temperature versus tropical water storage versus CGR in detrended anomalies. The values of CGR are indicated by the color-bar. **b.** Histograms of climate-CGR interannual correlations in the first three decades (1960-1989) and in the recent three decades (1989-2018), derived using 5000 bootstrapping repeats. Both tropical water storage (WS) and lagged precipitation (LagP) are used to represent tropical water availability. **c.** same as b, but showing histograms of the partial correlations of CGR to tropical temperature and tropical water after controlling tropical water and tropical temperature, respectively. **d.** Histograms of the interannual sensitivity of CGR to tropical WS (γ_{WS}) and LagP (γ_{LagP}) in the univariate regression for the same two period, derived using 5000 bootstrapping repeats. Unlike correlations, γ_{WS} and γ_{LagP} differ in magnitude due to the differences in WS and LagP IAV magnitude and are therefore shown separately. The unit of this sensitivity is Pg C year⁻¹ per Tt H₂O. **e.** same as d, but showing γ_{WS} and γ_{LagP} estimated using the bivariate regression with both tropical water and tropical temperature as predictors. Ridge regression is used here to reduce biases from high collinearity between water and temperature (Method).

We also utilize a 25-year moving window aiming to provide gradual changes of interannual sensitivity of CGR to water variations. As pointed out by the reviewer, γ_{WS} from the bivariate regression seem to stall starting from 1997 (1997 is the central year of the window of 1985-2009) (**Fig. R2-2**). However, these interannual changes of γ_{WS} after 1997 only reflect the very small changes of γ_{WS} at a 25-year period during the recent 34-year and are not apparent compared to the changes of γ_{WS} from the first 30-year to the recent 30-year. Also, γ_{WS} is not significantly different from 0 in the most prior 25-year windows but only become significantly negative in the recent 25-year windows ($P < 0.05$). Therefore, these small changes of γ_{WS} after 1997 do not change our main conclusions. Nonetheless, 25-year windows can show temporally gradual changes and it is useful to mention these small changes after 1997 in the main part of the manuscript, which we now do. We added the following text on this point (Lines 124-126):

“In addition, results based on a moving 25-year window show the sensitivity stalls in the recent 34-year period, i.e., after the time window centered on 1997(1985-2009) (Fig. S4). “

Fig. R2-2 Interannual sensitivity of CGR to tropical water availability variations based on 25-year windows from 1960 to 2018. a, CGR is the function of tropical temperature and tropical water storage. Sensitivity is estimated using the ridge regression. Each dot indicates a 25-year period. The central year of time window is labeled on the horizontal axis and two double-headed arrows are to clearly illustrate the coverage of the time windows. Shaded areas represent the 95% confidence interval, derived using 5000 bootstrapping repeats. b, same as a, but replacing tropical water storage with tropical lagged precipitation.

In summary, to address the Reviewer's comments, **we have included new comprehensive analyses into the revised MS and replace original Figure1 with Figure R2-1 and add a new Table R2-1**, which further increase the confidence in our findings and demonstrate their robustness. The finding that the interannual sensitivity and correlations of CGR to tropical water consistently became more negative from the prior 30-year to the recent past 30-year between 1960 and 2018 is robust.

References:

Hoerl, A. E., and R. W. Kennard. “Ridge Regression: Biased Estimation for Nonorthogonal Problems.” *Technometrics*. Vol. 12, No. 1, 1970, pp. 55–67.

T. Hastie, R. Tibshirani, J. Friedman, *The Elements of Statistical Learning: Data Mining, Inference, and Prediction*, Springer Series in Statistics (Springer, 2009).

Peng, S., Piao, S., Ciais, P., Myneni, R.B., Chen, A., Chevallier, F., Dolman, A.J., Janssens, I.A., Penuelas, J., Zhang, G. and Vicca, S., 2013. Asymmetric effects of daytime and night-time warming on Northern Hemisphere vegetation. *Nature*, 501(7465), pp.88-92.

Sippel, S., Meinshausen, N., Székely, E., Fischer, E., Pendergrass, A.G., Lehner, F. and Knutti, R., 2021. Robust detection of forced warming in the presence of potentially large climate variability. *Science advances*, 7(43), p.eabh4429.

C2.2:

2) Spatial coherence of water storage anomaly and ENSO. I appreciate that the authors have moved the correlation maps to the main figure 2c,d. However, I disagree that these maps show that the CP ENSO patterns explain the changing spatial coherence of the WS anomaly. The spatial coherence is not calculated in these maps. I agree with the authors' interpretation of Figure 2c,d that the

correlations with the water storage anomaly at each location are higher for the CP SSTs (panel d) than for the EP SSTs (panel c). But this is to be expected when we know that the tropically averaged water storage anomaly correlates well with the CP SSTs, which was already shown in panels a and b of figure 2. So with panels 2a-d you conclusively show that the water storage anomaly varies with CP. But this says nothing about how the spatial coherence is changing because of ENSO (the intriguing topic of panel 2e), and therefore it does not provide any support for the authors' claim that ENSO patterns might be affecting the change in the way that interannual water storage anomalies affect the CGR. This in turn leaves me wondering what Figures 2a-d add to the manuscript.

Response: We appreciate the reviewer's feedback in highlighting this point, which really helps us to improve the manuscript. **We now provide new evidence to demonstrate that ENSO can explain the increasing spatial coherence of water storage anomaly over time.**

To illustrate ENSO impacts on the spatial coherence more clearly, we now show year-to-year variations of spatial coherence and mark ENSO years (**Fig. R2-3a**). All years are classified into three groups according to the level of spatial coherence: low level (0th-33.3th percentile); medium level (33.3th-66.6th percentile); high level (66.6th-100th percentile). It is apparent that most years with high spatial coherence are ENSO years (**Fig. R2-3a and b**), but there is still a small proportion of high spatial coherence that cannot be explained by ENSO, suggesting the role of other factors, such as the tropical Atlantic variability (*De Linage et al., 2013; Rodell et al., 2018*). Compared to ENSO-neutral years, the spatial patterns of WS anomaly during ENSO years are quite uniform (**Fig. R2-4 and R2-5**).

From the first 30-year to the recent 30-year between 1960 and 2018, the share of years at high spatial coherence levels rose from 30% to 41%, mainly due to increased contributions of CP ENSO (**Fig. R2-3b**). The fact that CP dominates the ENSO types since the 1990s is well documented (*Ashok et al., 2007; Capotondi et al., 2015; Freund et al., 2019; Cai et al., 2021*). Consistent with our previous analyses, we further demonstrate that CP ENSO has a dominant impact on WS anomaly and spatial coherence during 1989-2018.

To further confirm the impacts of spatial coherence on tropical water-CGR coupling, we perform a new subset analysis (**Fig. R2-3c**) to complement the analysis of Fig. 2e in the previous MS. We detrend the data of each group based on spatial coherence levels and calculate correlations. **Fig. R2-3c** shows that $R(\text{WS}, \text{CGR})$ are only highly negative when spatial coherence of the tropical WS anomaly is high. $R(\text{WS}, \text{CGR} | T)$ confirms that this dependence of water-carbon coupling on the spatial coherence is not influenced by confounding temperature effects. This result again suggests that high spatial coherence of WS anomaly could result in highly negative tropical water-CGR coupling.

Therefore, ENSO-driven increasing spatial coherence over time could partly increase the terrestrial water-carbon coupling. **In the revised MS, we clarified this issue and use Fig. R2-3 to replace the original Fig. 2 (See Lines 162-189).** We also discuss other possible mechanisms in the revised MS (**See Lines 274-284**).

Fig. R2-3 ENSO-driven increasing spatial coherence of tropical WS anomaly impacts tropical water-CGR coupling during 1960-2018. **a.** Year-to-year variations of spatial coherence of WS anomaly and ENSO. All years are classified into three groups according to the level of spatial coherence: low level (0th-33.3th percentile); medium level (33.3th-66.6th percentile); high level (66.6th-100th percentile). Year was considered EP ENSO when the largest DJF SST anomaly over the region of 2°S-2°N, 110°E-90°W lies in the Eastern Pacific (Eastern of 150°W) and Nino3 index exceeds one standard deviation. Year was considered CP ENSO when the corresponding largest DJF SST anomaly lies in the Central Pacific (Western of 150°W) and Nino4 index exceeds one standard deviation. Volcano years are excluded from analyses. **b.** Share and component of years in high spatial coherence levels from the first 30-year (1960-1989) to the recent 30-year (1989-2018). Neutral years are identified as years not in the EP or CP ENSO state. **c.** Dependence of $R(WS,CGR)$ and $R(WS,CGR|T)$ on the spatial coherence of WS anomaly. The data of each group is detrended before calculating correlations. ** indicates a significant correlation at $P < 0.05$.

Fig. R2-4 Spatial pattern of WS anomaly at ENSO-neutral example years during 1960-2018.

Fig. R2-5 Spatial pattern of WS anomaly at EP and CP ENSO years during 1960-2018.

References:

de Linage, C., Kim, H., Famiglietti, J.S. and Yu, J.Y., 2013. Impact of Pacific and Atlantic sea surface temperatures on interannual and decadal variations of GRACE land water storage in tropical South America. *Journal of Geophysical Research: Atmospheres*, 118(19), pp.10-811.

Rodell, M., Famiglietti, J.S., Wiese, D.N., Reager, J.T., Beaudoing, H.K., Landerer, F.W. and Lo, M.H., 2018. Emerging trends in global freshwater availability. *Nature*, 557(7707), pp.651-659.

Ashok, K., Behera, S.K., Rao, S.A., Weng, H. and Yamagata, T., 2007. El Niño Modoki and its possible teleconnection. *Journal of Geophysical Research: Oceans*, 112(C11).

Capotondi, A., Wittenberg, A.T., Newman, M., Di Lorenzo, E., Yu, J.Y., Braconnot, P., Cole, J., Dewitte, B., Giese, B., Guilyardi, E. and Jin, F.F., 2015. Understanding ENSO diversity. *Bulletin of the American Meteorological Society*, 96(6), pp.921-938.

Freund, M.B., Henley, B.J., Karoly, D.J., McGregor, H.V., Abram, N.J. and Dommenges, D., 2019. Higher frequency of Central Pacific El Niño events in recent decades relative to past centuries. *Nature Geoscience*, 12(6), pp.450-455.

Cai, W., Santoso, A., Collins, M., Dewitte, B., Karamperidou, C., Kug, J.S., Lengaigne, M., McPhaden, M.J., Stuecker, M.F., Taschetto, A.S. and Timmermann, A., 2021. Changing El Niño–Southern Oscillation in a warming climate. *Nature Reviews Earth & Environment*, 2(9), pp.628-644.

Minor:

Figure S4: typo in label

Response: Corrected.

Reviewer #4 (Remarks to the Author):

I've been asked to review the author's replies to referee #1. I have read through the rebuttal letter as well as the manuscript and supplementary figures. Going through each comment and response, I can confirm that the authors have done a good job in responding to the comment and making the necessary changes in the paper.

Response: We thank the reviewer for the positive feedback.

Overall, the reviewer did not have any "fatal" concerns with the original manuscript. Rather, the reviewer wanted to see: 1) a better use of the model data, 2) a better clarification of what was meant by IAV and trend, and more 3) more clear take-home messages. Many of the point by point comments concern #1 and the authors have done well at addressing each of these issues regarding the use in models. While I agree with the reviewer that this kind of result (i.e., the models don't get it) is usually not very helpful, the authors have added additional figures and discussion concerning why the models might be wrong.

On #2 and #3, the main take home message of the paper is clear, i.e., that the coupling between tropical water availability and CGR appears (note that correlation and regressions used in the paper don't show causation so the "coupling" may be due to entirely different reasons) to have increased (trend) over the last 60 years for which we have data. The result that the carbon cycling in the tropics is also controlled by water availability has already been noted in several recent studies, but the result that this control has changed over the last 60 years possibly related to the spatial shifts in ENSO events seems to be the paper's novelty. It is up to the editor to decide whether that warrants a space in their prestigious publication.

Response: We thank the reviewer for the careful review and feedback. As highlighted in our response to reviewer #2, the increasing control of CGR by water variability and a contribution of changes in ENSO to this feature are both robust findings as supported by the additional provided analyses.

Reviewer #5 (Remarks to the Author):

Dear editor,

C5.0:

Because you said you are not asking for a full review, but to assess the authors' replies to the referee #3. The question of "Reviewer C3.2" is important. The datasets used in the author's main text show that the tropical water storage has comparable but opposite control on CGR in the 1960s and 2000s with the partial correlation coefficients about 0.4. Authors suggest that water control from 0.4 in 1960s to -0.4 in 2000s shows the increasing coupling between tropical carbon cycle and soil water, owing to the spatial coherence of WS anomaly. Referee#3 argued that if -0.4 in 2000s is owing to the increase in spatial coherence of WS, 0.4 in 1960s should also require the coherence in WS anomaly. In the reply, authors did not directly address whether coherence of WS can explain the 0.4 in 1960s based on their datasets in the main text. However, they used other datasets (GPCC, CRU precipitation) in the reply file to show that 0.4 cannot hold. So what's the reason of 0.4 in 2000s?

Response: We appreciate the Reviewer's feedback in highlighting this point again. We now provide new evidence to clarify the interpretation of \$R(\text{WS}, \text{CGR}|\text{T})\$.

First, according to the partial correlation equation (Eq. R5-1), $R(\text{WS}, \text{CGR}|\text{T})$ is determined by $R(\text{WS}, \text{CGR})$, $R(\text{WS}, \text{T})$ and $R(\text{T}, \text{CGR})$. We list estimates of each item in Table R5-1. The result shows

that $R(WS,CGR)$ becomes more negative from the first 25-year to the recent 25-year, which exactly supports our conclusions.

$$R(Water, Carbon|T) = \frac{R(Water, Carbon) - R(Water, T)R(T, Carbon)}{\sqrt{1 - R(Water, T)^2} \sqrt{1 - R(T, Carbon)^2}} \quad \text{Eq. R5-1}$$

Table R5-1. Tropical Climate-CGR coupling during 1960-1984 and 1994-2018

	1960-1984 (First 25-year period, three volcano years removed)	1994-2018 (Recent 25-year period, no volcano years)
R(WS,CGR)	-0.40*	-0.74**
R(T,CGR)	0.67**	0.77**
R(WS,T)	-0.83**	-0.77**
R(WS,CGR T)	0.37	-0.36*

Note: ** and * indicates a significant regression coefficient at $P < 0.05$ and $P < 0.1$, respectively. Volcano years are removed from analyses.

Second, after controlling the confounding water-temperature coupling, the specific values of partial correlation between WS and CGR become smaller, i.e., they become more positive, and their statistical significance all become lower for two sub-periods. **In other words, after controlling confounding water-temperature coupling, $R(WS,CGR|T)$ suggests that water impacts on CGR is much weaker: the probability that WS-CGR correlation is different from 0 is less than 90% at the prior 25-year and is between 90% and 95% at the recent past 25-year.**

In the previous MS, the interpretation of $R(W,CGR|T)$ is unclear and cause misunderstandings. Now, we clarify that $R(W,CGR|T)$ is not appropriate to be interpreted as total water impacts on CGR in the two periods. According to Eq. R5-1, $R(Water,CGR|T)$ isolates water impacts on CGR from confounding water-temperature coupling by linearly removing all temperature-related covariations. However, given the well-documented soil moisture-atmosphere feedbacks (*Seneviratne et al., 2010*), atmospheric temperature variability includes many feedbacks from soil moisture (e.g., hot extremes at tropical semi-arid regions) and removing all of them would indirectly remove some water impacts on CGR because of their physical connection. There is many observational and modelling evidence for these soil moisture-atmospheric feedbacks, as reviewed in *Seneviratne et al., 2010*. **This is exactly supported by a recent study (*Humphrey et al., 2021 Nature*). This study utilized model factorial experiments to demonstrate that removing soil moisture interannual variability suppresses land carbon uptake variability by about 90%, while tropical mean temperature remains unchanged (*Extended Data Fig. 10 in Humphrey et al., 2021 Nature*). Models do not reproduce the increasing water-carbon coupling but roughly capture the sign and strength of long-term tropical water-carbon coupling during 1960-2018, thus providing insights into underpinning processes.** Therefore, *Humphrey et al. 2021* suggested that tropical mean temperature might not represent a mechanistic climatic driver for land carbon uptake variability. Hence, $R(WS,CGR|T)$ could be an insufficient and less accurate measure to infer the sign and strength of independent water impacts on CGR and may underestimate total water impacts on CGR

in phases in which temperature control is dominant. This help explains why total water impacts on CGR, interpreted from $R(WS,CGR|T)$, are weak statistically during the two sub-periods.

Third, the confounding water-temperature coupling is very stable over time and cannot explain the increasing negative water-carbon coupling (**Table R5-1** and **Fig. R5-1**). Relative changes of $R(WS,CGR|T)$ directly help confirm that the increasing negative tropical water-carbon coupling remains robust. Furthermore, ENSO-driven increasing spatial coherence of water storage anomalies over time are plausible to partly explain the increasing $R(WS,CGR)$ and then $R(WS,CGR|T)$. However, since the process-based models do not reproduce this observed feature, the full elucidation of the underpinning mechanisms is difficult. **We also discuss other possible mechanisms, in addition to ENSO-driven increasing spatial coherence (See Lines 275-284).**

Fig. R5-1 Interannual tropical water-temperature correlations during 1989-2018 based on a 25-year moving window. Each dot indicates a 25-year period. The central year is labeled on the horizontal axis (for instance, 1975 represents the period of 1963-1987).

Fourth, the robust finding needs cross-validations from different data sets and approaches, and we make conclusions based on multiple lines of evidence. Therefore, **in combination the Reviewer2's points about the interannual sensitivity of CGR to terrestrial water, we perform a comprehensive analysis to construct the full picture of the evidence.**

Table R5-2 and **Figure R5-2** show that both interannual correlation and sensitivity of CGR to terrestrial water variations estimated from all approaches are significantly different between the two periods (based on the Wilcoxon signed-rank test, $P < 0.01$) and become more negative over time, despite the specific magnitude and unit of each metric depends on the chosen approach. For metrics considering the tropical mean temperature, the probability that they are different from 0 at the prior 30-year is low (26.8%) but high at the recent past 30-year (96.7%). For metrics without considering tropical mean temperature, metrics are significantly different from 0 at the same two periods ($P < 0.01$) and consistently become more negative. For instance, the univariate interannual sensitivity of CGR to tropical water increased by about 35% on average from the prior 30-year to the recent past 30-year.

These results robustly confirm that both sensitivity and correlation estimate consistently has become increasingly negative in the recent past (1989-2018) compared to prior climate conditions (1960-1989). Results based on a 25-year moving window are also added to in the revised Manuscript MS. We include all these results since they allow for a transparent discussion of uncertainties regarding confounding water-temperature coupling and do not affect our conclusions. In the revised Manuscript (MS), we now add estimates of sensitivity by replacing Figure R5-2 with original Figure1 and adding Table R5-2. We now also revise the MS title to “Increasing negative coupling between tropical water and interannual CO₂ growth rate” and other statements throughout the text.

Table R5-2. Summary of the interannual sensitivity and correlation of CGR to terrestrial water variations during the prior 30-year (1960-1989) and the recent past 30-year (1989-2018). Estimates are derived from 5000 bootstrapping repeats by randomly selecting years without volcano perturbations in each sub-period. The mean and one standard deviation are presented for the metric. The probability that the metric is different from 0 and the corresponding P value are computed by inverting the corresponding confidence intervals. For instance, the probability that the metric is different from 0 is 96% suggests that 96% confidence level of the metric does not include 0 and the corresponding P value is 0.04.

Metric	Method	Proxy of Water	Metrics during first 30-year	Probability that Metrics during first 30-year is different from 0	Metrics during recent 30-year	Probability that Metrics during recent 30-year is different from 0	Test whether the Means of Metrics during the two 30-year periods is equal (Wilcoxon signed-rank test)
Sensitivity	Univariate (OLS)	WS	-0.95 ± 0.27**	99.9%	-1.26 ± 0.23**	99.9%	Not, P<0.01
		LagP	-0.24 ± 0.07**	99.4%	-0.33 ± 0.05**	99.9%	Not, P<0.01
	Bivariate (Ridge)	WS	-0.05 ± 0.33	15.6%	-0.66 ± 0.26**	98.9%	Not, P<0.01
		LagP	-0.06 ± 0.1	43.4%	-0.22 ± 0.05**	99.9%	Not, P<0.01
	Bivariate (OLS)	WS	0.33 ± 0.55	43.4%	-0.68 ± 0.44*	90.4%	Not, P<0.01
		LagP	-0.03 ± 0.13	4.9%	-0.24 ± 0.07**	99.7%	Not, P<0.01
Correlation	R(W,CGR)	WS	-0.53 ± 0.14**	99.9%	-0.71 ± 0.1**	99.9%	Not, P<0.01
		LagP	-0.58 ± 0.15**	99.2%	-0.80 ± 0.06**	99.9%	Not, P<0.01
	R(W,CGR T)	WS	0.13 ± 0.21	44.5%	-0.36 ± 0.20*	90.4%	Not, P<0.01
		LagP	-0.04 ± 0.24	2.7%	-0.56 ± 0.14**	99.6%	Not, P<0.01
				55.3% (On average)		97.9% (On average)	

Note: ** and * indicate a significant sensitivity or correlation at P<0.05 and P<0.1, respectively.

Figure R5-2. Tropical land climate-carbon interannual relationships during 1960-2018. **a.** Yearly tropical temperature versus tropical water storage versus CGR in detrended anomalies. The values of CGR are indicated by the color-bar. **b.** Histograms of climate-CGR interannual correlations in the first three decades (1960-1989) and in the recent three decades (1989-2018), derived using 5000 bootstrapping repeats. Both tropical water storage (WS) and lagged precipitation (LagP) are used to represent tropical water availability. **c.** same as b, but showing histograms of the partial correlations of CGR to tropical temperature and tropical water after controlling tropical water and tropical temperature, respectively. **d.** Histograms of the interannual sensitivity of CGR to tropical WS (γ_{WS}) and LagP (γ_{LagP}) in the univariate regression for the same two period, derived using 5000 bootstrapping repeats. Unlike correlations, γ_{WS} and γ_{LagP} differ in magnitude due the differences in WS and LagP IAV magnitude and are therefore shown separately. The unit of this sensitivity is Pg C year^{-1} per $\text{Tt H}_2\text{O}$. **e.** same as d, but showing γ_{WS} and γ_{LagP} estimated using the bivariate regression with both tropical water and tropical temperature as predictors. Ridge regression is used here to reduce biases from high collinearity between water and temperature (Method).

In summary, we have provided new evidence to clarify the interpretation of $R(\text{WS},\text{CGR}|\text{T})$. **To avoid misunderstandings of $R(\text{Water},\text{CGR}|\text{T})$, as shown below, we also revised the main text (Also see Lines 90-99) and added a new section in Method to clarify the usefulness and weakness of partial correlation (Also see Lines 411-430 in revised MS).**

Lines 90-99:

“To check whether this increasing negative water-carbon correlation is influenced by the possible confounding water-temperature coupling, we look at temporal dynamics of water-temperature correlations and find they are very stable over time (Fig. S1). Partial correlations remove water-temperature correlations, and their relative changes directly help confirm that the increasing negative

tropical water-carbon coupling remains robust (Fig. 1c and Table S1). We note that since terrestrial water variability can also indirectly impact the land carbon cycle by triggering atmospheric temperature extremes through the well-documented soil moisture-atmosphere feedbacks^{22,23}, it could be not appropriate to interpret $R(W,CGR|T)$ as total water impacts on CGR in the two periods but the temporal changes of $R(W,CGR|T)$ are useful here (Methods).

Lines 406-425:

“Partial correlation. Partial correlation is used here to directly check whether increased water-carbon coupling is influenced by confounding water-temperature coupling. However, using specific values of $R(W,CGR|T)$ to conclude the sign and strength of total water impacts on CGR is not suggested. $R(W,CGR|T)$ isolates water impacts on CGR from confounding water-temperature coupling by linearly removing all temperature-related covariations. However, given the well-documented soil moisture-atmosphere feedbacks²³, temperature variability actually includes many feedbacks from soil moisture (e.g., hot extremes at tropical semi-arid regions) and removing all of them would indirectly remove some water impacts on CGR because of their physical connection. In addition, models do not reproduce the increasing water-carbon coupling but roughly capture the sign and strength of long-term tropical water-carbon coupling during 1960-2018, thus providing insights into underpinning processes. Model factorial experiments show that removing soil moisture interannual variability suppresses land carbon uptake variability by about 90%, while tropical mean temperature remains unchanged (Extended Data Fig. 10 in ref. ²²). Therefore, ref. ²² suggest that tropical mean temperature might not represent a mechanistic climatic driver for land carbon uptake variability. Hence, $R(W,CGR|T)$ is an insufficient and less accurate measure to infer the sign and strength of independent water impacts on CGR and underestimate water impacts on CGR in phases in which temperature control is dominant. Nonetheless, their relative changes are useful in this study and the finding has become increasingly negative in the recent past (1989-2018) compared to prior climate conditions (1960-1989) is robust.”

References:

Seneviratne, S. I., Corti, T., Davin, E. L., Hirschi, M., Jaeger, E. B., Lehner, I., ... & Teuling, A. J. (2010). Investigating soil moisture–climate interactions in a changing climate: A review. *Earth-Science Reviews*, 99(3-4), 125-161.

Humphrey, V., Berg, A., Ciais, P., Gentile, P., Jung, M., Reichstein, M., Seneviratne, S.I. and Frankenberg, C., 2021. Soil moisture–atmosphere feedback dominates land carbon uptake variability. *Nature*, 592(7852), pp.65-69.

C5.1:

Another concern is that these partial correlation coefficients were calculated in a 25-yr moving window. In Figure 1d (main text), a lot of correlation coefficients ranging from 0.4 in 1960s to -0.4 in 2000s have no statistical significance (close to zero). Even that they use GPCC/CRU in Table R3-1 to calculate the partial correlation coefficients, all the correlation coefficients in the first 30-yr are not significant.

In the t-test for the correlation and regression, it needs the degree of freedom (dof). The moving window will cause high auto-correlation, making the dof low (so dof is not equal to N-2). They need to re-calculate the dof to calculate the significance (not just the p-values directly given by python or R functions).

Response: We appreciate the reviewer's feedback in highlighting this point.

As the reviewer mentioned and the new **Table R5-2** and **Figure R5-2**, R(Water,CGR|T) is close to 0 in the first 30-year but only becomes highly negative in the recent 30-year between 1960 and 2018. This result exactly supports our conclusions: The interannual relationship between tropical terrestrial water availability and CGR has become increasingly negative in the recent past (1989-2018) compared to prior climate conditions (1960-1989).

We thank the reviewer for pointing out the issue of autocorrelations from the moving window. We follow the method recommended in *Chelton (1984)* to recalculate effective sample size and associated dof:

$$\frac{1}{N_{eff}} = \frac{1}{N} + \frac{2}{N} \sum_{j=1}^m \rho_X(j)\rho_Y(j) \quad \text{Eq. R5-2}$$

where N_{eff} is the effective sample size; N is the original sample size, $\rho_X(j)$ and $\rho_Y(j)$ are the autocorrelations of X and Y at lag j , respectively. We did a cutoff at the length of the moving window m to focus on the artificial autocorrelation due to the moving average. The p-value is then derived from the standard t-test. Note that *Pyper & Peterman (1998)* compared a series of different methods for computing the effective sample size, and the one described in *Chelton (1984)* gives a lower error rate than other methods. With a much smaller effective sample size, the p-values rise yet are significant at the 0.05 level. For instance, for the linear trend of R(WS,CGR) from 25-year moving windows, p-value changes from 4.3841e-15 to 0.0054 after taking the autocorrelations into account.

The use of a 25-year moving window is aiming to show temporally gradual changes of terrestrial climate-carbon coupling at a 25-year period between 1960 and 2018. However, these gradual changes are not necessarily in a linear way, to avoid confusions, we delete these linear fittings in the revised MS. The comparison of the first 30-year period and the recent 30-year robustly supports our conclusion.

References:

Chelton, D.B. 1984. Commentary: short-term climatic variability in the Northeast Pacific Ocean. The influence of ocean conditions on the production of salmonids in the North Pacific. Edited by W. Pearcy. Oregon State University Press, Corvallis, Oreg. pp. 87–99.

Pyper, B.J. and Peterman, R.M., 1998. Comparison of methods to account for autocorrelation in correlation analyses of fish data. Canadian Journal of Fisheries and Aquatic Sciences, 55(9), pp.2127-2140.

Reviewer Reports on the Second Revision:

Referees' comments:

Referee #2 (Remarks to the Author):

I thank the authors for their revisions. These revisions have addressed my concern about the sensitivity vs. correlation-based results. I remain somewhat concerned about the patterns in Figure S4 but I appreciate that these late years are probably calculated from uneven moving windows, and I believe the other changes the authors have made have partially strengthened the argument. I also agree the revised Figure 2 better shows that the spatial coherence seems to shift with ENSO patterns. I should note that this was quite difficult to make up from the Figure, though - I had to squint at the years one by one with pen and paper in Figure 2a to really see that the high sensitivity years are more common in the ENSO group, and the axis in Figure 2b is a bit confusing. The grouping into low, medium, and high sensitivity is also fairly arbitrary. It may be cleaner to just show a scatter plot of spatial coherence vs. the Nino3 index. This is up to the authors to decide though and affects only ease of understanding.

Referee #5 (Remarks to the Author):

After modifications, this manuscript has been improved, with more reliable statistical methods and conclusions. It reveals that tropical water availability is very likely increasingly controlling the interannual variability of terrestrial carbon cycle and might continue to dominate tropical terrestrial carbon-climate feedbacks. Some comments as follows:

(1) L164-166: suggesting the role of other factors, like the tropical Atlantic variability. Maybe include Indian Ocean Dipole (IOD). Recently they have made some investigations on the effect of IOD on tropical terrestrial land carbon cycle, such as "Wang, J., et al. (2022), Enhanced India-Africa Carbon Uptake and Asia-Pacific Carbon Release Associated With the 2019 Extreme Positive Indian Ocean Dipole, *Geophys Res Lett*, 49(22)." And Wang, J., et al. (2021), Modulation of Land Photosynthesis by the Indian Ocean Dipole: Satellite-Based Observations and CMIP6 Future Projections, *Earth's Future*, 9(4)."

(2) L175: "We detrend the data of each group based on the spatial coherence level before calculating correlations and find that $R(\text{WS}, \text{CGR})$ is only highly negative when the spatial coherence of the tropical WS anomaly is high (Fig. 2c)". I have a little confusion about why you did not pick out the anomalies from the long-term CGR anomaly, but you detrend the subset data again. The anomalies from these two methods can be different.

(3) In the EP and CP, they actually take different effects on variations of CGR and terrestrial carbon cycle, which has been revealed by Chylek et al. (2018) and Wang et al. (2018). They may provide some information.

Chylek, P., P. Tans, J. Christy, and M. K. Dubey (2018), The carbon cycle response to two El Nino types: an observational study, *Environmental Research Letters*, 13(2).

Wang, J., et al. (2018), Contrasting interannual atmospheric CO₂ variabilities and their terrestrial

mechanisms for two types of El Ninos, Atmos. Chem. Phys., 18, 10333-10345.

(4) In the section "Diagnosis of CMIP6 models". You evaluate 9 ESMs and 6 LSMs, and pointed out that they do not reproduce the intensification of water-carbon coupling. One major concern is that the coupled models have some issues in simulating ENSO events, especially the probabilities of EP and CP occurrences in historical run, which can largely mislead the results. So in coupled simulation, the performance of ENSO events, adjustments of circulations in each model, precipitation diagnosis, and process representations of the carbon cycle's response to soil water etc. can influence their results. Therefore, it is not only the potential lack of some critical process representations (L233-234). In my opinion, these coupled ESMs cannot provide further implications here.

(5) Additionally, you analyzed the NEP in this part. Actually, CGR interannual variability is more related to NBP. Specifically, carbon flux from the wildfires maybe take effects in the intensifying water-carbon coupling, as you revealed here. As I know, many models provide NBP in CMIP6, maybe you can check which models include the fire simulations.

(6) L403: you use the sum of soil moisture in all layers to represent terrestrial water storage.

Different models have different layers and depths, making the sum of soil moisture largely different. Is it possible to use soil moisture in the basically same depths?

Author Rebuttals to Second Revision:

We thank the editor and the reviewers for their insightful comments and suggestions. The manuscript has been revised following the reviewers' comments and editorial suggestions. Please see our detailed response below. The original reviewer comments are displayed in black, and our responses are in **blue**.

Reviewer' comments:

Reviewer #2 (Remarks to the Author):

I thank the authors for their revisions. These revisions have addressed my concern about the sensitivity vs. correlation-based results. I remain somewhat concerned about the patterns in Figure S4 but I appreciate that these late years are probably calculated from uneven moving windows, and I believe the other changes the authors have made have partially strengthened the argument. I also agree the revised Figure 2 better shows that the spatial coherence seems to shift with ENSO patterns. I should note that this was quite difficult to make up from the Figure, though - I had to squint at the years one by one with pen and paper in Figure 2a to really see that the high sensitivity years are more common in the ENSO group, and the axis in Figure 2b is a bit confusing. The grouping into low, medium, and high sensitivity is also fairly arbitrary. It may be cleaner to just show a scatter plot of spatial coherence vs. the Nino3 index. This is up to the authors to decide though and affects only ease of understanding.

Response: We appreciate the reviewer's acknowledgment that our revisions have addressed concerns. To improve the ease of understanding, we now add vertical lines connecting the symbols of high spatial coherence and ENSO and modify the Y-axis label of Figure 2b to "Fraction (%) and ENSO state of high spatial coherence years".

Reviewer #5 (Remarks to the Author):

After modifications, this manuscript has been improved, with more reliable statistical methods and conclusions. It reveals that tropical water availability is very likely increasingly controlling the interannual variability of terrestrial carbon cycle and might continue to dominate tropical terrestrial carbon-climate feedbacks. Some comments as follows:

Response: We appreciate the reviewer's acknowledgment that the conclusions are robust. We now address the remaining points below.

(1) L164-166: suggesting the role of other factors, like the tropical Atlantic variability. Maybe include Indian Ocean Dipole (IOD). Recently they have made some investigations on the effect of IOD on tropical terrestrial land carbon cycle, such as "Wang, J., et al. (2022), Enhanced India-Africa Carbon Uptake and Asia-Pacific Carbon Release Associated With the 2019 Extreme Positive Indian Ocean Dipole, *Geophys Res Lett*, 49(22)." And Wang, J., et al. (2021), Modulation of Land Photosynthesis by the Indian Ocean Dipole: Satellite-Based Observations and CMIP6 Future Projections, *Earth's Future*,9(4)."

Response: We now mention the role of Indian Ocean Dipole and cite the suggested two papers. [Line 148]

(2) L175: “We detrend the data of each group based on the spatial coherence level before calculating correlations and find that $R(\text{WS}, \text{CGR})$ is only highly negative when the spatial coherence of the tropical WS anomaly is high (Fig. 2c)”. I have a little confusion about why you did not pick out the anomalies from the long-term CGR anomaly, but you detrend the subset data again. The anomalies from these two methods can be different.

Response: We thank the reviewer for pointing this out. We now utilize the suggested method. The result of water-carbon coupling dependence on spatial coherence remains almost unchanged because the interannual variations of CGR/Climate anomalies differ slightly between the two methods. We now clarify it in the revised MS.

Lines 157-159:

” We first detrend all years of data by removing the long-term trend and then bin them into three subsets according to spatial coherence levels. $R(\text{WS}, \text{CGR})$ is highly negative only when the spatial coherence is high (Fig. 2c)”.

(3) In the EP and CP, they actually take different effects on variations of CGR and terrestrial carbon cycle, which has been revealed by Chylek et al. (2018) and Wang et al. (2018). They may provide some information.

Chylek, P., P. Tans, J. Christy, and M. K. Dubey (2018), The carbon cycle response to two El Nino types: an observational study, *Environmental Research Letters*, 13(2).

Wang, J., et al. (2018), Contrasting interannual atmospheric CO₂ variabilities and their terrestrial mechanisms for two types of El Ninos, *Atmos. Chem. Phys.*, 18, 10333-10345.

Response: We now highlight the view that EP and CP could take different impacts on terrestrial carbon cycle and suggest further investigations and cite the suggested two papers. [Lines 170-172]

(4) In the section “Diagnosis of CMIP6 models”. You evaluate 9 ESMs and 6 LSMs, and pointed out that they do not reproduce the intensification of water-carbon coupling. One major concern is that the coupled models have some issues in simulating ENSO events, especially the probabilities of EP and CP occurrences in historical run, which can largely mislead the results. So in coupled simulation, the performance of ENSO events, adjustments of circulations in each model, precipitation diagnosis, and process representations of the carbon cycle's response to soil water etc. can influence their results. Therefore, it is not only the potential lack of some critical process representations (L233-234). In my opinion, these coupled ESMs cannot provide further implications here.

Response: Corrected. We now clarify the implication of coupled ESMs in the revised MS.

Lines 198-200:

“For coupled ESMs, the underpinning reason is more complex, for instance, they have known issues in simulating the probability of occurrence of historical EP ENSO and CP ENSO [Timmermann et al., 2018].”

Reference:

Timmermann, A. et al. El Nino-Southern Oscillation complexity. Nature 559, 535-545 (2018).

(5) Additionally, you analyzed the NEP in this part. Actually, CGR interannual variability is more related to NBP. Specifically, carbon flux from the wildfires maybe take effects in the intensifying water-carbon coupling, as you revealed here. As I know, many models provide NBP in CMIP6, maybe you can check which models include the fire simulations.

Response: We thank the reviewer for pointing this out. Following your suggestion, we additionally utilize the output of NBP from CMIP6, which include carbon fluxes from fire and other disturbances. As expected, using modeled NBP also cannot reproduce the intensified water-carbon coupling because NEE dominates the interannual variability of NBP in models. We now add it in the revised MS.

Lines 202-204:

“Further including modeled carbon fluxes from fire and other disturbances, i.e., replacing NEE with net biome production, cannot help explain the failure of models to reproduce the intensified water-carbon coupling (Supplementary Fig. 12)”.

(6) L403: you use the sum of soil moisture in all layers to represent terrestrial water storage. Different models have different layers and depths, making the sum of soil moisture largely different. Is it possible to use soil moisture in the basically same depths?

Response: We thank the reviewer for raising this question. Since we aim for a fair comparison of the water-carbon relationship between observations and models, similar to *Humphrey et al., 2018 Nature*, using modeled terrestrial water storage is more reasonable in this case. We now further explain this in the revised MS:

Lines 565-568:

“Following previous efforts [Humphrey et al., 2018; Wu et al., 2021], to enable a fair comparison of the water-carbon relationship between observations and models, we use the sum of soil moisture in all layers and snow water equivalent as modeled terrestrial water storage. In the tropics, snow water equivalent is negligible.”

References:

Humphrey, V., Zscheischler, J., Ciais, P., Gudmundsson, L., Sitch, S., & Seneviratne, S. I. (2018). Sensitivity of atmospheric CO₂ growth rate to observed changes in terrestrial water storage. Nature, 560(7720), 628-631.

Wu, R. J., Lo, M. H., & Scanlon, B. R. (2021). The annual cycle of terrestrial water storage anomalies in CMIP6 models evaluated against GRACE data. Journal of Climate, 34(20), 8205-8217.